# Confident Approximate Policy Iteration for Efficient Local Planning in $q^\pi$-realizable MDPs

**Gellért Weisz**
DeepMind, London, UK
University College London, London, UK

**András György**
DeepMind, London, UK

**Tadashi Kozuno**
University of Alberta, Edmonton, Canada
Omron Sinic X, Tokyo, Japan

**Csaba Szepesvári**
DeepMind, London, UK
University of Alberta, Edmonton, Canada

## Abstract

We consider approximate dynamic programming in $\gamma$-discounted Markov decision processes and apply it to approximate planning with linear value-function approximation. Our first contribution is a new variant of APPROXIMATE POLICY ITERATION (API), called CONFIDENT APPROXIMATE POLICY ITERATION (CAPI), which computes a deterministic stationary policy with an optimal error bound scaling linearly with the product of the effective horizon $H$ and the worst-case approximation error $\varepsilon$ of the action-value functions of stationary policies. This improvement over API (whose error scales with $H^2$) comes at the price of an $H$-fold increase in memory cost. Unlike Scherrer and Lesner [2012], who recommended computing a non-stationary policy to achieve a similar improvement (with the same memory overhead), we are able to stick to stationary policies. This allows for our second contribution, the application of CAPI to planning with local access to a simulator and $d$-dimensional linear function approximation. As such, we design a planning algorithm that applies CAPI to obtain a sequence of policies with successively refined accuracies on a dynamically evolving set of states. The algorithm outputs an $\tilde{O}(\sqrt{d}H\varepsilon)$-optimal policy after issuing $\tilde{O}(dH^4/\varepsilon^2)$ queries to the simulator, simultaneously achieving the optimal accuracy bound and the best known query complexity bound, while earlier algorithms in the literature achieve only one of them. This query complexity is shown to be tight in all parameters except $H$. These improvements come at the expense of a mild (polynomial) increase in memory and computational costs of both the algorithm and its output policy.

## 1 Introduction

A key question in reinforcement learning is how to use value-function approximation to arrive at scaleable algorithms that can find near-optimal policies in Markov decision processes (MDPs). A flurry of recent results aims at solving this problem efficiently with varying models of interaction with the MDP. In this paper we focus on the problem of *planning with a simulator* when using linear function approximation. A simulator is a "device" that, given a state-action pair as a query, returns a next state and reward generated from the transition kernel of the MDP that is simulated. Depending on the application, such a simulator is often readily available (e.g., in chess, go, Atari). Planning with simulator access comes with great benefits: for example, in a recent work, Wang et al. [2021] showed that under some conditions it is exponentially more efficient to find a near-optimal policy if a simulator of the MDP (that can reset to a state) is available compared to the online case where a learner interacts with its environment by following trajectories but without the help of a simulator.

36th Conference on Neural Information Processing Systems (NeurIPS 2022).

Our setting of *offline, local planning* considers the problem of finding a policy with near-optimal value at a given initial state $s_0$ in the MDP. The planner can issue queries to the simulator, and has to find and output a near-optimal policy with high probability. The efficiency of a planner is measured in four ways: the *suboptimality* of the policy found, that is, how far its value is from that of the optimal policy; the *query cost*, that is, the number of queries issued to the simulator; the *computational cost*, which is the number of operations used; and the *memory cost*, which is the amount of memory used (we adopt the real computation model for these costs). There are several interaction models between the planner and the simulator [Yin et al., 2022]. The most permissive one is called the *generative model*, or *random access*. Here, the planner receives the set of all states and is allowed to issue queries for any state and action. Coding a simulator that supports this model can be challenging, as oftentimes the set of states is computationally difficult to describe. Instead of *random access*, we consider the more practical and more challenging *local access* setting, where the planner only sees the initial state and the set of states received as a result to a query to the simulator. Consequently, the queries issued have to be for a state that has already been encountered this way (and any available action), while the simulator needs to support the ability to reset the MDP state *only* to previously seen states. A simple approach in practice to support this model is saving and reloading checkpoints during the operation of the simulator.

To handle large, possibly infinite state spaces, we use linear function approximation to approximate the action-value functions $q^\pi$ of stationary, deterministic policies $\pi$ (for background on MDPs, see the next section). A feature-map is a good fit to an MDP if the worst-case error of using the feature-map to approximate value functions of policies of the MDP is small:

**Definition 1.1** ($q^\pi$-realizability: uniform policy value-function approximation error). *Given an MDP, the uniform policy value-function approximation error induced by a feature map $\varphi$, which maps state-action pairs $(s, a)$ to the Euclidean ball of radius $L$ centered at zero in $\mathbb{R}^d$, over a set of parameters belonging to the d-dimensional centered Euclidean ball of radius $B$ is*

$$\varepsilon = \sup_{\pi} \inf_{\theta: \|\theta\|_2 \le B} \sup_{(s,a)} |q^\pi(s, a) - \langle \varphi(s, a), \theta \rangle|,$$

*where the outermost supremum is over all possible stationary deterministic memoryless policies (i.e., maps from states to actions) of the MDP.*

Our goal is to design algorithms that scale gracefully with the uniform approximation error $\varepsilon$ at the expense of controlled computational cost. To achieve nontrivial guarantees, the uniform approximation error needs to be "small". This (implicit) assumption is stronger than the $q^\star$-realizability assumption (where the approximation error is only considered for optimal policies), which Weisz et al. [2021] showed an exponential query complexity lower bound for. At the same time, it is (strictly) weaker than the linear MDP assumption [Zanette et al., 2020], for which there are efficient algorithms to find a near-optimal policy in the online setting (without a simulator) [Jin et al., 2020], even in the more challenging reward-free setting where the rewards are only revealed after exploration [Wagenmaker et al., 2022].

In the *local access* setting, the planner learns the features $\varphi(s, a)$ of a state-action pair *only* for those states $s$ that have already been encountered. In contrast, in the *random access* setting, the whole feature map $\varphi(\cdot, \cdot)$, of (possibly infinite) size $d|\mathcal{S}||\mathcal{A}|$ (where $\mathcal{S}$ and $\mathcal{A}$ are the state and action sets, resp.), is given to the planner as input. In the latter setting, when only the query cost is counted, Du et al. [2019] and Lattimore et al. [2020] proposed algorithms (the latter working in the misspecified, $\varepsilon > 0$ regime) that issue a number of queries that is polynomial in the relevant parameters, but require a barycentric spanner or near-optimal design of the input features. In the worst case, computing any of these sets scales polynomially in $|\mathcal{S}|$ and $|\mathcal{A}|$, which can be prohibitive.

In the case of *local access*, considered in this paper, the best known bound on the suboptimality of the computed policy is achieved by CONFIDENT MC-POLITEX [Yin et al., 2022]. In the more permissive *random access* setting, the best known query cost is achieved by Lattimore et al. [2020]. Our algorithm, CAPI-QPI-PLAN (given in Algorithm 3), achieves the *best of both* while only assuming *local access*. This is shown in the next theorem; in the theorem $\varepsilon$ is as defined in Definition 1.1, $\gamma$ is the discount factor, and $v^\star$ and $v^\pi$ are the state value functions associated with the optimal policy and policy $\pi$, respectively (precise definitions of these quantities are given in the next section). A comparison to other algorithms in the literature is given in Table 1; there the accuracy parameter $\omega$ of the algorithms is set to $\varepsilon$, but a larger $\omega$ can be used to trade off suboptimality guarantees for an improved query cost.

**Theorem 1.2.** *For any confidence parameter $\delta \in (0, 1]$, accuracy parameter $\omega > 0$, and initial state $s_0 \in \mathcal{S}$, with probability at least $1 - \delta$, CAPI-QPI-PLAN (Algorithm 3) finds a policy $\pi$ with*

$$v^\star(s_0) - v^\pi(s_0) = \tilde{O}\left((\varepsilon + \omega)\sqrt{d}(1 - \gamma)^{-1}\right), \tag{1}$$

*while executing at most $\tilde{O}\left(d(1 - \gamma)^{-4}\omega^{-2}\right)$ queries in the* local access *setting.*

CAPI-QPI-PLAN is based on CONFIDENT MC-LSPI, another algorithm of Yin et al. [2022], which relies on policy iteration from a *core set* of informative state-action pairs, but achieves inferior performance both in terms of suboptimality and query complexity. However, CAPI-QPI-PLAN's improvements come at the expense of increased memory and computational costs, as shown in the next theorem: compared to CONFIDENT MC-LSPI, the memory and computational costs of our algorithm increase by a factor of the effective horizon $H = \tilde{O}(1/(1 - \gamma))$, and the policy computed by CAPI-QPI-PLAN uses a $dH$ factor more memory for storage and a $d^2H$ factor more computation to evaluate.

**Theorem 1.3** (Memory and computational cost)**.** *The memory and computational cost of running* CAPI-QPI-PLAN *(Algorithm 3) are $\tilde{O}\left(d^2/(1 - \gamma)\right)$ and $\tilde{O}\left(d^4|\mathcal{A}|(1 - \gamma)^{-5}\omega^{-2}\right)$, respectively, while the memory and computational costs of storing and evaluating the final policy outputted by* CAPI-QPI-PLAN, *respectively, are $\tilde{O}\left(d^2/(1 - \gamma)\right)$ and $\tilde{O}\left(d^3|\mathcal{A}|/(1 - \gamma)\right)$.*

Next we present a lower bound corresponding to Theorem 1.2 that holds even in the more permissive *random access* setting, and shows that CAPI-QPI-PLAN trades of the query cost and the suboptimality of the returned policy asymptotically optimally up to its dependence on $1/(1 - \gamma)$:

**Theorem 1.4** (Query cost lower bound)**.** *Let $\alpha \in (0, \frac{0.05\gamma}{(1-\gamma)(1+\gamma)^2})$, $\delta \in (0, 0.08]$, $\gamma \in [\frac{7}{12}, 1]$, $d \geq 3$, and $\varepsilon \geq 0$. Then there is a class $\mathcal{M}$ of MDPs with uniform policy value-function approximation error at most $\varepsilon$ such that any planner that guarantees to find an $\alpha$-optimal policy $\pi$ (i.e., $v^\star(s_0) - v^\pi(s_0) \leq \alpha$) with probability at least $1 - \delta$ for all $M \in \mathcal{M}$ when used with a simulator for $M$ with* random access, *the worst-case (over $\mathcal{M}$) expected number of queries issued by the planner is at least*

$$\max\left(\exp\left(\Omega\left(\frac{d\varepsilon^2}{\alpha^2(1 - \gamma)^2}\right)\right), \Omega\left(\frac{d^2}{\alpha^2(1 - \gamma)^3}\right)\right). \tag{2}$$

If $\omega$ is set to $\varepsilon$ for CAPI-QPI-PLAN, the first term of Eq. (2) implies that any planner with an asymptotically smaller (apart from logarithmic factors) suboptimality guarantee than Eq. (1) executes exponentially many queries in expectation. The second term of Eq. (2), which is shown to be a lower bound in Theorem H.3 even in the more general setting of linear MDPs with zero misspecification ($\varepsilon = 0$), matches the query complexity of Theorem 1.2 up to an $\tilde{O}((1 - \gamma)^2)$ factor. Thus, the lower bound implies that the suboptimality and query cost bounds of Theorem 1.2 are tight up to logarithmic factors in all parameters except the $1/(1 - \gamma)$-dependence of the query cost bound.

At the heart of our method is a new algorithm, which we call CONFIDENT APPROXIMATE POLICY ITERATION (CAPI). This algorithm, which belongs to the family of approximate dynamic programming algorithms [Bertsekas, 2012, Munos, 2003, 2005], is a novel variant of APPROXIMATE POLICY ITERATION (API) [Bertsekas and Tsitsiklis, 1996]: in the policy improvement step, CAPI only updates the policy in states where it is confident that the update will improve the performance. This simple modification allows CAPI to avoid the problem of "classical" approximate dynamic programming algorithms (approximate policy and value iteration) of inflating the value function evaluation error by a factor of $H^2$ where $H = \tilde{O}(1/(1 - \gamma))$ (for discussions of this problem, see also the papers by Scherrer and Lesner, 2012 and Russo, 2020), and reduce this inflation factor to $H$. A similar result has already been achieved by Scherrer and Lesner [2012], who proposed to construct a non-stationary policy that strings together all policies obtained while running either approximate value or policy iteration. However, applying this result to our planning problem is problematic, since the policies to be evaluated are non-stationary, and hence including them in the policy set we aim to approximate may drastically increase the error $\varepsilon$ as compared to Definition 1.1, which only considers stationary memoryless policies.

While the improvements provided by CAPI allows CAPI-QPI-PLAN to match the performance of CONFIDENT MC-POLITEX in terms of suboptimality, it is unlikely that a simple modification of CONFIDENT MC-POLITEX would lead to an algorithm which matches CAPI-QPI-PLAN's

**Table 1:** Comparison of suboptimality and query complexity guarantees of various planners (with the approximation accuracy parameter $\omega$ set to $\varepsilon$). Drawbacks are highlighted with red, the best bounds with blue.

| Algorithm [Publication] | Query cost | Suboptimality | Access model |
|---|---|---|---|
| MC-LSPI [Lattimore et al., 2020] | $\tilde{O}\left(\frac{d}{\varepsilon^2(1-\gamma)^4}\right)$ | $\tilde{O}\left(\frac{\varepsilon\sqrt{d}}{(1-\gamma)^2}\right)$ | random access |
| CONFIDENT MC-LSPI [Yin et al., 2022] | $\tilde{O}\left(\frac{d^2}{\varepsilon^2(1-\gamma)^4}\right)$ | $\tilde{O}\left(\frac{\varepsilon\sqrt{d}}{(1-\gamma)^2}\right)$ | local access |
| CONFIDENT MC-POLITEX [Yin et al., 2022] | $\tilde{O}\left(\frac{d}{\varepsilon^4(1-\gamma)^5}\right)$ | $\tilde{O}\left(\frac{\varepsilon\sqrt{d}}{1-\gamma}\right)$ | local access |
| CAPI-QPI-PLAN [This work] | $\tilde{O}\left(\frac{d}{\varepsilon^2(1-\gamma)^4}\right)$ | $\tilde{O}\left(\frac{\varepsilon\sqrt{d}}{1-\gamma}\right)$ | local access |

performance in terms of query cost (see Table 1): Both methods evaluate a sequence of policies at an $\tilde{O}(\varepsilon)$ accuracy each (requiring $\tilde{O}(1/\varepsilon^2)$ queries, omitting the dependence on other parameters). However, while CAPI-QPI-PLAN (and CONFIDENT MC-LSPI) evaluates $O(\log(1/\varepsilon))$ (again in terms of $\varepsilon$ only) policies to find one which is $\tilde{O}(\varepsilon)$-optimal, CONFIDENT MC-POLITEX needs to compute $\tilde{O}(1/\varepsilon^2)$ policies to achieve the same. As a consequence, CONFIDENT MC-POLITEX only achieves $\tilde{O}(1/\varepsilon^4)$ query complexity, and to match CAPI-QPI-PLAN's $\tilde{O}(1/\varepsilon^2)$ complexity, one would need to come up with either significantly better policy evaluation methods (potentially using the similarity in the subsequent policies) or a much faster (exponential vs. square-root) convergence rate in the suboptimality of the policy sequence produced by CONFIDENT MC-POLITEX.

The rest of the paper is organized as follows: The model and notation are introduced in Section 2. CAPI is introduced and analyzed in Section 3. Planning with $q^\pi$-realizability is introduced in Section 4, with CAPI-QPI-PLAN being built-up and analyzed in Sections 4.1 and 4.2. In particular, the proof of Theorem 1.2 is given in Section 4.2. Several proofs are relegated to appendices, in particular, Theorem 1.3 is proved and implementation details of CAPI-QPI-PLAN are discussed in Appendix G, while Theorem 1.4 is proved in Appendix H.

## 2 Notation and preliminaries

Let $\mathbb{N} = \{0, 1, \ldots\}$ denote the set of natural numbers, $\mathbb{N}^+ = \{1, 2, \ldots\}$ the positive integers. For some integer $i$, let $[i] = \{0, \ldots, i-1\}$. For $x \in \mathbb{R}$, let $\lceil x \rceil$ denote the smallest integer i such that $i \geq x$. For a positive definite $V \in \mathbb{R}^{d \times d}$ and $x \in \mathbb{R}^d$, let $\|x\|_V^2 = x^\top V x$. For matrices $A$ and $B$, we say that $A \geq B$ if $A - B$ is positive semidefinite. Let $\mathbb{I}$ be the $d$-dimensional identity matrix. For compatible vectors $x, y$, let $\langle x, y \rangle$ be their inner product: $\langle x, y \rangle = x^\top y$. Let $\mathcal{M}_1(X)$ denote the space of probability distributions supported on the set $X$ (throughout, we assume that the $\sigma$-algebra is implicit). We write $a \approx_\varepsilon b$ for $a, b, \varepsilon \in \mathbb{R}$ if $|a - b| \leq \varepsilon$. We denote by $\tilde{O}(\cdot)$ and $\tilde{\Theta}(\cdot)$ the variants of the big-O notation that hide polylogarithmic factors.

A Markov Decision Process (MDP) is a tuple $M = (\mathcal{S}, \mathcal{A}, Q)$, where $\mathcal{S}$ is a measurable state space, $\mathcal{A}$ is a finite action space, and $Q : \mathcal{S} \times \mathcal{A} \to \mathcal{M}_1(\mathcal{S} \times [0, 1])$ is the transition-reward kernel. We define the transition and reward distributions $P : \mathcal{S} \times \mathcal{A} \to \mathcal{M}_1(\mathcal{S})$ and $\mathcal{R} : \mathcal{S} \times \mathcal{A} \to \mathcal{M}_1([0, 1])$ as the marginals of $Q$. By a slight abuse of notation, for any $s \in \mathcal{S}$ and $a \in \mathcal{A}$, let $P(\cdot|s, a)$ and $\mathcal{R}(\cdot|s, a)$ denote the distributions $P(s, a)$ and $\mathcal{R}(s, a)$, respectively. We further denote by $r(s, a) = \int_0^1 x \, d\mathcal{R}(x|s, a)$ the expected reward for an action $a \in \mathcal{A}$ taken in a state $s \in \mathcal{S}$. Without loss of generality, we assume that there is a designated initial state $s_0 \in \mathcal{S}$.

Starting from any state $s \in \mathcal{S}$, a stationary memoryless policy $\pi : \mathcal{S} \to \mathcal{M}_1(\mathcal{A})$ interacts with the MDP in a sequential manner for time-steps $t \in \mathbb{N}$, defining a probability distribution $\mathcal{P}_{\pi,s}$ over the episode trajectory $\{S_i, A_i, R_i\}_{i \in \mathbb{N}}$ as follows: $S_0 = s$ deterministically, $A_i \sim \pi(S_i)$, and $(S_{i+1}, R_i) \sim Q(S_i, A_i)$. By a slight variation, let $\mathcal{P}_{\pi,s,a}$ denote (for some $a \in \mathcal{A}$) the distribution of the trajectory when $A_0 = a$ deterministically, while the distribution of the rest of the trajectory is defined analogously.

This allows us to conveniently define the expected state-value and action-value functions in the discounted setting we consider, for some discount factor $0 < \gamma < 1$, respectively, as

$$v^\pi(s) = \mathbb{E}_{\pi,s}\left[\sum_{t \in \mathbb{N}} \gamma^t R_t\right] \quad \text{and} \quad q^\pi(s, a) = \mathbb{E}_{\pi,s,a}\left[\sum_{t \in \mathbb{N}} \gamma^t R_t\right] \qquad \text{for all } (s, a) \in \mathcal{S} \times \mathcal{A}, \quad (3)$$

where throughout the paper we use the convention that $\mathbb{E}_{\bullet}$ is the expectation operator corresponding to a distribution $\mathcal{P}_{\bullet}$ (e.g., $\mathbb{E}_{\pi,s}$ is the expectation with respect to $\mathcal{P}_{\pi,s}$). It is well known (see, e.g., Puterman, 1994) that there exists an optimal stationary deterministic memoryless policy $\pi^{\star}$ such that

$$\sup_{\pi} v^{\pi}(s) = v^{\pi^{\star}}(s) \qquad \text{and} \qquad \sup_{\pi} q^{\pi}(s, a) = q^{\pi^{\star}}(s, a) \qquad \text{for all } (s, a) \in \mathcal{S} \times \mathcal{A}.$$

Let $v^{\star} = v^{\pi^{\star}}$ and $q^{\star} = q^{\pi^{\star}}$. For any policy $\pi$, $v^{\pi}$ and $q^{\pi}$ are known to satisfy the Bellman equations [Puterman, 1994]:

$$v^{\pi}(s) = \sum_{a \in \mathcal{A}} \pi(a|s) q^{\pi}(s, a) \text{ and } q^{\pi}(s, a) = r(s, a) + \gamma \int_{s' \in \mathcal{S}} v^{\pi}(s') \, dP(s'|s, a) \text{ for all } (s, a) \in \mathcal{S} \times \mathcal{A}. \quad (4)$$

Finally, we call a policy $\pi$ deterministic if for all states, $\pi(s)$ is a distribution that assigns unit weight to one action and zero weight to the others. With a slight abuse of notation, for a deterministic policy $\pi$, we denote by $\pi(s)$ the action $\pi$ chooses (deterministically) in state $s \in \mathcal{S}$.

# 3   Confident Approximate Policy Iteration

In this section we introduce CONFIDENT APPROXIMATE POLICY ITERATION (CAPI), our new approximate dynamic programming algorithm. In approximate dynamic programming, the methods are designed around oracles that return either an approximation to the application of the Bellman optimality operator to a value function ("approximate value iteration"), or an approximation to the value function of some policy ("approximate policy iteration"). Our setting is the second. The novelty is that we assume access to the accuracy of the approximation and use this knowledge to modify the policy update, which leads to improved guarantees on the suboptimality of the computed policy.

We present the pseudocodes of API [Bertsekas and Tsitsiklis, 1996] and CAPI jointly in Algorithm 1: starting from an arbitrary (deterministic) policy $\pi_0$, the algorithm iterates a policy estimation (Line 2) and a policy update step (Line 3) $I$ times. The policy update for API is greedy with respect to the action-value estimates $\hat{q}$ and is defined as $\pi_{\hat{q}}(s) = \arg\max_{a \in \mathcal{A}} \hat{q}(s, a)$. We assume that $\arg\max_{a \in \mathcal{A}}$ breaks ties in a consistent manner by ordering the actions (using the notation $\mathcal{A} = (\mathcal{A}_1, \ldots, \mathcal{A}_{|\mathcal{A}|})$) and always choosing action $\mathcal{A}_i$ with the lowest index $i$ that achieves the maximum. For CAPI, the policy update further relies on a global estimation-accuracy parameter $\omega$, and a set of fixed-states $\mathcal{S}_{\text{fix}} \subseteq \mathcal{S}$. For the purposes of this section, it is enough to keep $\mathcal{S}_{\text{fix}} = \{\}$. CAPI updates the policy to one that acts greedily with respect to $\hat{q}$ *only* on states that are not in $\mathcal{S}_{\text{fix}}$ and where it is confident that this leads to an improvement over the previous policy (Case 5a); otherwise, the new policy will return the same action as the previous one (Case 5b).   To decide, $\hat{q}(s, \pi(s)) + \omega$ is treated as the upper bound on the previous policy's value, and $\max_{a \in \mathcal{A}} \hat{q}(s, a) - \omega$ as the lower bound of the action-value of the greedy action (Eq. 5):

$$\pi_{\hat{q}, \pi, \mathcal{S}_{\text{fix}}}(s) = \begin{cases} \arg\max_{a \in \mathcal{A}} \hat{q}(s, a), & \text{if } s \notin \mathcal{S}_{\text{fix}} \text{ and } \hat{q}(s, \pi(s)) + \omega < \max_{a \in \mathcal{A}} \hat{q}(s, a) - \omega; & (5a) \\ \pi(s), & \text{otherwise.} & (5b) \end{cases}$$

Note that $\pi_{\hat{q}, \pi, \mathcal{S}_{\text{fix}}}$ also depends on $\omega$, however, this dependence is omitted from the notation (as $\omega$ is kept fixed throughout).

CAPI can also be seen as a refinement of CONSERVATIVE POLICY ITERATION (CPI) of Kakade and Langford [2002] with some important differences: While CPI introduces a global parameter to ensure the update stays close to the previous policy, CAPI has no such parameter, and it dynamically decides when to stay close to (more precisely, use) the previous policy, individually for every state, based on whether there is evidence for a guaranteed improvement.

Let $\pi$ be any stationary deterministic memoryless policy, $\hat{q}^{\pi} : \mathcal{S} \times \mathcal{A} \to \mathbb{R}$ be any function, $\omega \in \mathbb{R}_{+}$, and $\mathcal{S}_{\text{fix}} \subseteq \mathcal{S}$. First, we show that as long as $\hat{q}^{\pi}$ is an $\omega$-accurate estimate of $q^{\pi}$, the CAPI policy update only improves the policy's values:

**Lemma 3.1** (No deterioration). *Let $\pi' = \pi_{\hat{q}^{\pi}, \pi, \mathcal{S}_{\text{fix}}}$. Assume that for all $s \in \mathcal{S} \setminus \mathcal{S}_{\text{fix}}$ and $a \in \mathcal{A}$, $\hat{q}^{\pi}(s, a) \approx_{\omega} q^{\pi}(s, a)$. Then, for any $s \in \mathcal{S}$, $v^{\pi'}(s) \geq v^{\pi}(s)$.*

*Proof.* Fix any $s \in \mathcal{S}$. If $s \in \mathcal{S}_{\text{fix}}$ or $\hat{q}^{\pi}(s, \pi(s)) + \omega \geq \max_{a \in \mathcal{A}} \hat{q}^{\pi}(s, a) - \omega$, then $\pi'(s) = \pi(s)$ and therefore $q^{\pi}(s, \pi'(s)) = v^{\pi}(s)$. Otherwise, $s \notin \mathcal{S}_{\text{fix}}$ and $\hat{q}^{\pi}(s, \pi(s)) + \omega \leq \max_{a \in \mathcal{A}} \hat{q}^{\pi}(s, a) - \omega$, hence $\pi'(s) = \arg\max_{a \in \mathcal{A}} \hat{q}^{\pi}(s, a)$, and it follows by our assumptions that $q^{\pi}(s, \pi'(s)) \geq$

---

**Algorithm 1** APPROXIMATE POLICY ITERATION (API) and CONFIDENT APPROXIMATE POLICY ITERATION (CAPI)

---
1: **for** $i = 1$ to $I$ **do**
2: $\quad \hat{q} \leftarrow \text{ESTIMATE}(\pi_{i-1})$
3: $\quad \pi_i \leftarrow \begin{cases} \pi_{\hat{q}} & \text{API} \\ \pi_{\hat{q}, \pi_{i-1}, S_{\text{fix}}} & \text{CAPI} \end{cases}$
4: **return** $\pi_I$

---

$\hat{q}^\pi(s, \pi'(s)) - \omega = \max_{a \in \mathcal{A}} \hat{q}^\pi(s, a) - \omega > \hat{q}^\pi(s, \pi(s)) + \omega \geq q^\pi(s, \pi(s)) = v^\pi(s)$. Therefore, in any case, $q^\pi(s, \pi'(s)) \geq v^\pi(s)$. Since this holds for any $s \in \mathcal{S}$, the Policy Improvement Theorem [Sutton and Barto, 2018, Section 4.2] implies that for any $s \in \mathcal{S}$, $v^{\pi'}(s) \geq v^\pi(s)$. □

Next we introduce two approximate optimality criterion for a policy on a set of states:

**Definition 3.2** (Policy optimality on a set of states). *A policy $\pi$ is $\Delta$-optimal (for some $\Delta \geq 0$) on a set of states $\mathcal{S}' \subseteq \mathcal{S}$, if for all $s \in \mathcal{S}'$, $v^\star(s) - v^\pi(s) \leq \Delta$.*

**Definition 3.3** (Next-state optimality on a set of states). *A policy $\pi$ is next-state $\Delta$-optimal (for some $\Delta \geq 0$) on a set of states $\mathcal{S}' \subseteq \mathcal{S}$, if for all $s \in \mathcal{S}'$ and all actions $a \in \mathcal{A}$, $\int_{s' \in \mathcal{S}} (v^\star(s') - v^\pi(s')) \, dP(s'|s, a) \leq \Delta$.*

Note that in the special case of $\mathcal{S}' = \mathcal{S}$ the first property implies the second, that is, if $\pi$ is $\Delta$-optimal on $\mathcal{S}$, then it is also next-state $\Delta$-optimal on $\mathcal{S}$. Next, we show that the suboptimality of a policy updated by CAPI evolves as follows (the proof is relegated to Appendix A):

**Lemma 3.4** (Iteration progress). *Let $\pi' = \pi_{\hat{q}^\pi, \pi, S_{\text{fix}}}$. Assume that for all $s \in \mathcal{S} \setminus \mathcal{S}_{\text{fix}}$ and $a \in \mathcal{A}$, $\hat{q}^\pi(s, a) \approx_\omega q^\pi(s, a)$, and that $\pi$ is next-state $\Delta$-optimal on $\mathcal{S} \setminus \mathcal{S}_{\text{fix}}$. Then $\pi'$ is $(4\omega + \gamma\Delta)$-optimal on $\mathcal{S} \setminus \mathcal{S}_{\text{fix}}$.*

### 3.1 CAPI guarantee with accurate estimation everywhere

To obtain a final suboptimality guarantee for CAPI, first consider the ideal scenario in which we assume that we have a mechanism to estimate $q^\pi(s, a)$ up to some $\omega$ accuracy for all $s \in \mathcal{S}$ and $a \in \mathcal{A}$, and for any policy $\pi$:

**Assumption 3.5.** *There is an oracle called* ESTIMATE *that accepts a policy $\pi$ and returns $\hat{q}^\pi : \mathcal{S} \times \mathcal{A} \to \mathbb{R}$ such that for all $s \in \mathcal{S}$ and $a \in \mathcal{A}$, $\hat{q}^\pi(s, a) \approx_\omega q^\pi(s, a)$.*

**Theorem 3.6** (CAPI performance). *Assume CAPI (Algorithm 1) is run with $\mathcal{S}_{\text{fix}} = \{\}$, iteration count to $I = \lceil \log \omega / \log \gamma \rceil$, and suppose that the estimation used in Line 2 satisfies Assumption 3.5. Then the policy $\pi_I$ returned by the algorithm is $5\omega/(1 - \gamma)$-optimal on $\mathcal{S}$.*

*Proof.* We prove by induction that policy $\pi_i$ is $\Delta_i$-optimal on $\mathcal{S}$ for $\Delta_i = 4\omega \sum_{j \in [i]} \gamma^j + \frac{\gamma^i}{1-\gamma}$. This holds immediately for the base case of $i = 0$, as rewards are bounded in $[0, 1]$ and thus $v^\star(s) \leq 1/(1-\gamma)$ for any $s$. Assuming now that the inductive hypothesis holds for $i - 1$ we observe that $\pi_{i-1}$ is next-state $\Delta$-optimal on $\mathcal{S} = \mathcal{S} \setminus \mathcal{S}_{\text{fix}}$. Together with Assumption 3.5, this implies that the conditions of Lemma 3.4 are satisfied for $\pi = \pi_{i-1}$, which yields $v^\star(s) - v^{\pi_i}(s) \leq 4\omega + \gamma\Delta_{i-1} = \Delta_i$, finishing the induction. Finally, by the definition of $I$, $\pi_I$ is $\Delta_I$-optimal with $\Delta_I \leq \frac{4\omega}{1-\gamma} + \frac{\gamma^I}{1-\gamma} \leq \frac{5\omega}{1-\gamma}$. □

## 4 Local access planning with $q^\pi$-realizability

Our planner, CAPI-QPI-PLAN, is based on the CONFIDENT MC-LSPI algorithm of Yin et al. [2022]. This latter algorithm gradually builds a *core set* of state-action pairs whose corresponding features are informative. The $q$-values of the state-action pairs in the core set are estimated using rollouts. The procedure is restarted with an extended core set whenever the algorithm encounters a new informative feature. If such a new feature is not encountered, the estimation error can be controlled, and the estimation is extended to all state-action pairs using the least-squares estimator. Finally, the extended estimation is used in Line 2 of API.

CAPI-QPI-PLAN improves upon CONFIDENT MC-LSPI in two ways. First, using CAPI instead of API improves the final suboptimality bound by a factor of the effective horizon. Second, we

---

**Algorithm 2** MEASURE

---

1: **Input:** state $s$, action $a$, deterministic policy $\pi$, set of states $\mathcal{S}' \subseteq \mathcal{S}$, accuracy $\omega > 0$, failure probability $\zeta \in (0, 1]$
2: **Initialize:** $H \leftarrow \lceil \log((\omega/4)(1-\gamma))/\log\gamma \rceil$, $n \leftarrow \lceil (\omega/4)^{-2}(1-\gamma)^{-2}\log(2/\zeta)/2 \rceil$
3: **for** $i = 1$ to $n$ **do**
4: $\quad (S, R_{i,0}) \leftarrow$ SIMULATOR$(s, a)$
5: $\quad$ **for** $h = 1$ to $H - 1$ **do**
6: $\quad\quad$ **if** $S \notin \mathcal{S}'$ **then return** (discover, $S$)
7: $\quad\quad A \leftarrow \pi(S)$
8: $\quad\quad (S, R_{i,h}) \leftarrow$ SIMULATOR$(S, A)$ $\qquad\qquad\qquad$ ▷ Call to the simulator oracle
9: **return** (success, $\frac{1}{n} \sum_{i=1}^{n} \sum_{h=0}^{H-1} \gamma^h R_{i,h}$)

---

apply a novel analysis on a more modular variant of the CONFIDENTROLLOUT subroutine used in CONFIDENT MC-LSPI, which delivers $q$-estimation accuracy guarantees with respect to a large class of policies simultaneously. This allows for a dynamically evolving version of policy iteration, that does not have to restart whenever a new informative feature is encountered. Intuitively, this prevents duplication of work.

### 4.1 Estimation oracle

To obtain an algorithm for planning with local access whose performance degrades gracefully with the uniform approximation error, we must weaken Assumption 3.5. This is because under local access, we cannot guarantee to cover all states or hope to obtain accurate $q$-value estimates for all states. Instead, we are interested in an accuracy guarantee that holds for $q$-values only on some subset $\mathcal{S}' \subseteq \mathcal{S}$ of states, but holds simultaneously for *any* policy that agrees with $\pi$ on $\mathcal{S}'$ but may take arbitrary values elsewhere. For this, we define the extended set of policies:

**Definition 4.1.** *Let* $\Pi_{det}$ *be the set of all stationary deterministic memoryless policies,* $\pi \in \Pi_{det}$, *and* $\mathcal{S}' \subseteq \mathcal{S}$. *For* $(\pi, \mathcal{S}')$, *we define* $\Pi_{\pi, \mathcal{S}'}$ *to be the set of policies that agree with* $\pi$ *on* $s \in \mathcal{S}'$:

$$\Pi_{\pi, \mathcal{S}'} = \{\pi' \in \Pi_{det} : \pi(s) = \pi'(s) \text{ for all } s \in \mathcal{S}'\} \ .$$

We aim to first accurately estimate $q^\pi(s, a)$ for *some specific* $(s, a)$ pairs, based on which we extend the estimates to other state-action pairs using least-squares. To this end, we first devise a subroutine called MEASURE (Algorithm 2). MEASURE is a modularized variant of the CONFIDENTROLLOUT subroutine of Yin et al. [2022]. The modularity of our variant is due to the parameter $\mathcal{S}'$ that corresponds to the set of states on which the planner is "confident" for CONFIDENTROLLOUT. MEASURE unrolls the policy $\pi$ starting from $(s, a)$ for a number of episodes, each lasting $H$ steps, and returns with the average measured reward. Throughout, we let $H = \lceil \log((\omega/4)(1-\gamma))/\log\gamma \rceil$ be the effective horizon. At the end of this process, MEASURE returns status *success* along with the empirical average $q$-value, where compared to Eq. (3), the discounted summation of rewards is truncated to $H$. If, however, the algorithm encounters a state not in its input $\mathcal{S}'$, it returns with status *discover*, along with that state. This is because in such cases, the algorithm could no longer guarantee an accurate estimation with respect to any member of the extended set of policies. The next lemma, proved in Appendix B, shows that MEASURE provides accurate estimates of the action-value functions for members of the extended policy set.

**Lemma 4.2.** *For any input parameters* $s \in \mathcal{S}, a \in \mathcal{A}, \pi \in \Pi_{det}, \mathcal{S}' \subset \mathcal{S}, \omega > 0, \zeta \in (0, 1)$, MEASURE *either returns with* (discover, $s'$) *for some* $s' \notin \mathcal{S}'$ *(Line 6), or it returns with* (success, $\tilde{q}$) *such that with probability at least* $1 - \zeta$,

$$q^{\pi'}(s, a) \approx_\omega \tilde{q} \quad \text{for all} \quad \pi' \in \Pi_{\pi, \mathcal{S}'}. \tag{6}$$

Suppose we have a list of state-action pairs $C = (s_i, a_i)_{i \in [|C|]}$ and corresponding $q$-estimates $\bar{q} = (\bar{q}_i)_{i \in |C|}$. We use the regularized least-squares estimator LSE (Eq. 8) to extend the estimates for all state-action pairs, with regularization parameter $\lambda = \omega^2/B^2$ (recall that $B$ is defined in Definition 1.1):

$$V(C) = \lambda\mathbb{I} + \sum_{i \in [|C|]} \varphi(s_i, a_i)\varphi(s_i, a_i)^\top, \tag{7}$$

$$\text{LSE}_{C, \bar{q}}(s, a) = \left\langle \varphi(s, a), V(C)^{-1} \sum_{i \in [|C|]} \varphi(s_i, a_i)\bar{q}_i \right\rangle. \tag{8}$$

For $C = \bar{q} = ()$ (the empty sequence), we define $\text{LSE}_{C,\bar{q}}(\cdot,\cdot) = 0$. This estimator satisfies the guarantee below.

**Lemma 4.3.** *Let $\pi$ be a stationary deterministic memoryless policy. Let $C = (s_i, a_i)_{i \in [n]}$ be sequences of state-action pairs of some length $n \in \mathbb{N}$ and $\bar{q} = (\bar{q}_i)_{i \in [n]}$ a sequence of corresponding reals such that for all $i \in [n]$, $q^\pi(s_i, a_i) \approx_\omega \bar{q}_i$. Then, for all $s, a \in \mathcal{S} \times \mathcal{A}$,*

$$\left| LSE_{C,\bar{q}}(s,a) - q^\pi(s,a) \right| \le \varepsilon + \|\varphi(s,a)\|_{V(C)^{-1}} \left( \sqrt{\lambda}B + (\omega + \varepsilon)\sqrt{n} \right), \tag{9}$$

*where $\varepsilon$ is the uniform approximation error from Definition 1.1.*

The proof is given in Appendix C. The order of the estimation accuracy bound (Eq. 9) is optimal, as shown by the lower bounds of Du et al. [2019] and Lattimore et al. [2020].

We intend to use the LSE estimator given in Eq. (8) and the bound in Lemma 4.3 only for state-action pairs where $\|\varphi(s,a)\|_{V(C)^{-1}} \le 1$ (and $n = \tilde{O}(d)$). We call these state-action pairs *covered* by $C$, and we call a state $s$ covered by $C$ if for all their corresponding actions $a$, the pair $(s,a)$ is covered by $C$:

$$\text{ActionCover}(C) = \{(s,a) \in \mathcal{S} \times \mathcal{A} : \|\varphi(s,a)\|_{V(C)^{-1}} \le 1\} \tag{10}$$

$$\text{Cover}(C) = \{s \in \mathcal{S} : \forall a \in \mathcal{A}, (s,a) \in \text{ActionCover}(C)\}. \tag{11}$$

We will use the parameter $\mathcal{S}_{\text{fix}}$ of CAPI (see CAPI's update rule in Eq. 5) to ensure policies are only updated on covered states, where the approximation error is well-controlled by Eq. (9).

## 4.2  Main algorithm

Finally, we are ready to introduce CAPI-QPI-PLAN, presented in Algorithm 3, our algorithm for planning with local access under approximate $q^\pi$-realizability. For this, we define levels $l = 0, 1, \dots, H$, and corresponding suboptimality requirements: For any $l \in [H+1]$, let

$$\Delta_l = 8(\varepsilon + \omega)\left(\sqrt{\tilde{d}} + 1\right) \sum_{j \in [l]} \gamma^j + \frac{\gamma^l}{1 - \gamma},$$

for some $\tilde{d} = \tilde{\Theta}(d)$ defined in Eq. (13). For each level $l$, the algorithm maintains a policy $\pi_l$ and a set of covered states on which it can guarantee that $\pi_l$ is a $\Delta_l$-optimal policy. More specifically, this set is $\text{Cover}(C_l)$, where $C_l$ is a list of state-action pairs with elements $C_{l,i} = (s_l^i, a_l^i)$ for $i \in [|C_l|]$. The algorithm maintains the following suboptimality guarantee below, which we prove in Appendix E after showing some further key properties of the algorithm.

**Lemma 4.4.** *Assuming that Eq. (6) holds whenever MEASURE returns success, $\pi_l$ is $\Delta_l$-optimal on $\text{Cover}(C_l)$ (Definition 3.2) for all $l \in [H+1]$ at the end of every iteration of the main loop of* CAPI-QPI-PLAN.

CAPI-QPI-PLAN aims to improve the policies, while *propagating* the members of $C_l$ to $C_{l+1}$, and so on, all the way to $C_H$. During this, whenever the algorithm discovers a state-action pair with a sufficiently "new" feature direction, this pair is appended to the sequence $C_0$ corresponding to level 0, as there are no suboptimality guarantees yet available for such a state. However, such a discovery can only happen $\tilde{O}(d)$ times. When, eventually, all discovered state-action pairs end up in $C_H$, the final suboptimality guarantee is reached, and the algorithm returns with the final policy. Note that in the local access setting we consider, the algorithm cannot enumerate the set $\text{Cover}(C_l)$, but can answer membership queries, that is, for any $s \in \mathcal{S}$ it encounters, it is able to decide if $s \in \text{Cover}(C_l)$. The algorithm maintains sequences $\bar{q}_l$, corresponding to $C_l$, for each level $l$. Whenever a new $(s,a)$ pair is appended to the sequence $C_l$, a corresponding $\perp$ symbol is appended to the sequence $\bar{q}_l$, to signal that an estimate of $q^{\pi_l}(s,a)$ is not yet known.

After initializing $C_0$ to cover the initial state $s_0$ (Lines 4 to 6), the algorithm measures $q^{\pi_\ell}(s,a)$ for the smallest level $\ell$ for which there still exists a $\perp$ in the corresponding $\bar{q}_\ell$. After a successful measurement, if there are no more $\perp$'s left at this level (i.e., in $\bar{q}_\ell$), the algorithm executes a policy update on $\pi_\ell$ (Line 17) using the least-squares estimate obtained from the measurements at this level, but only for states in $\text{Cover}(C_\ell)$ (using $\mathcal{S}_{\text{fix}} = \mathcal{S} \setminus \text{Cover}(C_\ell)$). Next, Line 18 merges this new policy $\pi'$ with the existing policy $\pi_{\ell+1}$ of the next level, setting $\pi_{\ell+1}$ to be the policy $\pi''$ defined as

$$\pi''(s) = \begin{cases} \pi_{\ell+1}(s), & \text{if } s \in \text{Cover}(C_{\ell+1}); \\ \pi'(s), & \text{otherwise.} \end{cases}$$

---

**Algorithm 3** CAPI-QPI-PLAN

---

1: **Input:** initial state $s_0 \in \mathcal{S}$, dimensionality $d$, parameter bound $B$, accuracy $\omega$, failure probability
$\quad$ $\delta > 0$
2: **Initialize:** $H \leftarrow \lceil \log((\omega/4)(1-\gamma))/\log\gamma \rceil$, for $l \in [H+1]$, $C_l \leftarrow ()$, $\bar{q}_l \leftarrow ()$, $\pi_l \leftarrow$
$\quad$ policy that always returns action $\mathcal{A}_1$, $\lambda \leftarrow \omega^2/B^2$
3: **while** True **do** $\qquad\qquad\qquad\qquad\qquad\qquad\qquad\qquad\qquad\qquad\qquad\qquad\qquad$ ▷ main loop
4: $\quad$ **if** $\exists a \in \mathcal{A}$, $(s_0, a) \notin \text{ActionCover}(C_0)$ **then**
5: $\qquad$ append $(s_0, a)$ to $C_0$, append $\bot$ to $\bar{q}_0$
6: $\qquad$ **break**
7: $\quad$ let $\ell$ be the smallest integer such that $\bar{q}_{\ell,m} = \bot$ for some $m$; set $\ell = H$ if no such $l$ exists
8: $\quad$ **if** $\ell = H$ **then return** $\pi_H$
9: $\quad$ $(\text{status}, \text{result}) \leftarrow \text{MEASURE}(s_\ell^m, a_\ell^m, \pi_\ell, \text{Cover}(C_\ell), \omega, \delta/(\tilde{d}H))$ $\quad$ ▷ recall $C_{\ell,m} = (s_\ell^m, a_\ell^m)$
10: $\quad$ **if** status = discover **then**
11: $\qquad$ append $(\text{result}, a)$ to $C_0$ for some $a$ such that $(\text{result}, a) \notin \text{ActionCover}(C_0)$
12: $\qquad$ append $\bot$ to $\bar{q}_0$
13: $\qquad$ **break**
14: $\quad$ $\bar{q}_{\ell,m} \leftarrow \text{result}$
15: $\quad$ **if** $\nexists m'$ such that $\bar{q}_{\ell,m'} = \bot$ **then**
16: $\qquad$ $\hat{q} \leftarrow \text{LSE}_{C_\ell, \bar{q}_\ell}$
17: $\qquad$ $\pi' \leftarrow \pi_{\hat{q}, \pi_\ell, \mathcal{S} \setminus \text{Cover}(C_\ell)}$
18: $\qquad$ $\pi_{\ell+1} \leftarrow (s \mapsto \pi_{\ell+1}(s) \text{ if } s \in \text{Cover}(C_{\ell+1}) \text{ else } \pi'(s))$
19: $\qquad$ **for** $(s, a) \in C_\ell$ such that $(s, a) \notin C_{\ell+1}$ **do**
20: $\qquad\qquad$ append $(s, a)$ to $C_{\ell+1}$, $\bot$ to $\bar{q}_{\ell+1}$

---

This ensures that the existing policy $\pi_{\ell+1}$ remains unchanged by $\pi''$ (its replacement) on states that are already covered by $C_{\ell+1}$, and therefore $\pi'' \in \Pi_{\pi_{\ell+1}, \text{Cover}(C_{\ell+1})} = \Pi_{\pi'', \text{Cover}(C_{\ell+1})}$. We also observe that $C_l$ can only grow for any $l$ (elements are never removed from these sequences), thus for any update where $C_l$ is assigned a new value $C_l'$ (Lines 5, 11, and 20), $V(C_l') \succeq V(C_l)$, and therefore $\text{Cover}(C_l') \supseteq \text{Cover}(C_l)$ and $\Pi_{\pi_l, \text{Cover}(C_l')} \subseteq \Pi_{\pi_l, \text{Cover}(C_l)}$. Combining these properties yields the following result:

**Lemma 4.5.** *If for any $l \in [H]$, $\pi_l$ and $C_l$ take some values $\pi_l^{old}$ and $C_l^{old}$ at any point in the execution of the algorithm, then at any later point during the execution, $\pi_l \in \Pi_{\pi_l, \text{Cover}(C_l)} \subseteq \Pi_{\pi_l^{old}, \text{Cover}(C_l^{old})}$.*

Any value in $\bar{q}_l$ that is set to anything other than $\bot$ will never change again. Since as long as the sample paths generated by MEASURE in Line 9 of CAPI-QPI-PLAN remain in $\text{Cover}(C_l)$, their distribution is the same under any policy from $\Pi_{\pi_l, \text{Cover}(C_l)}$, the $\bar{q}_l$ estimates are valid for these policies, as well. Combined with Lemma 4.5, we get that the accuracy guarantees of Lemma 4.2 continue to hold throughout:

**Lemma 4.6.** *Assuming that Eq. (6) holds whenever MEASURE returns success, for any level $l$ and index $m$ such that $\bar{q}_{l,m} \neq \bot$, $q^{\pi'}(s_l^m, a_l^m) \approx_\omega \bar{q}_{l,m}$ for all $\pi' \in \Pi_{\pi_l, \text{Cover}(C_l)}$ throughout the execution of CAPI-QPI-PLAN.*

Once $\pi_{\ell+1}$ is updated in Line 18, in Line 20 we append to the sequence $C_{\ell+1}$ all members of $C_\ell$ that are not yet in $C_{\ell+1}$, while adding a corresponding $\bot$ to $\bar{q}_{\ell+1}$ indicating that these $q$-values are not yet measured for policy $\pi_{\ell+1}$. Thus, whenever all $\bot$ values disappear from some level $l \in [H+1]$, by the end of that iteration $C_{l+1} = C_l$, and hence $\text{ActionCover}(C_l) = \text{ActionCover}(C_{l+1})$. Together with the fact that for any $l \in [H+1]$, whenever a new state-action pair is appended to $C_l$, an $\bot$ symbol is appended to $\bar{q}_l$, we have by induction the following result:

**Lemma 4.7.** *Throughout the execution of CAPI-QPI-PLAN, after Line 7 when $\ell$ is set,*

$$\text{ActionCover}(C_0) = \text{ActionCover}(C_1) = \cdots = \text{ActionCover}(C_\ell).$$

As a result, whenever the MEASURE call of Line 9 outputs $(\text{discover}, s)$ for some state $s$, by Lemma 4.2, there is an action $a \in \mathcal{A}$ such that $(s, a) \notin \text{ActionCover}(C_\ell) = \text{ActionCover}(C_0)$. This explains why adding such an $(s, a)$ pair to $C_0$ is always possible in Line 11. Consider the $i^{\text{th}}$ time Line 11 is executed, and denote $s$ by $s_i$ and $a$ by $a_i$, and $V_i = \lambda \mathbb{I} + \sum_{t=1}^{i-1} \varphi(s_t, a_t)\varphi(s_t, a_t)^\top$. Observe that as $V_i = V(C)$, $(s_i, a_i) \notin \text{ActionCover}(C_0)$ implies $\|\varphi(s_i, a_i)\|_{V_i^{-1}} > 1$. Therefore,

$\sum_{t=1}^{i} \min\{1, \|\varphi(s_t, a_t)\|_{V_t^{-1}}\} = i$, and thus by the elliptical potential lemma [Lattimore and Szepesvári, 2020, Lemma 19.4], $i \leq 2d \log\left(\frac{d\lambda + iL^2}{d\lambda}\right)$. This inequality is satisfied by the largest value of $i$, that is, the total number of times MEASURE returns with *discover*. Since any element of $C_l$ is also an element of $C_0$ for any $l \in [H+1]$, we have that at any time during the execution of CAPI-QPI-PLAN,

$$|C_l| \leq 4d \log\left(1 + \frac{4L^2}{\lambda}\right) =: \tilde{d} = \tilde{O}(d). \tag{13}$$

When CAPI-QPI-PLAN returns at Line 8 with the policy $\pi_H$, it is $\Delta_H$-optimal on $\text{Cover}(C_H)$ by Lemma 4.4 when the estimates of MEASURE are correct. Furthermore, $s_0 \in \text{Cover}(C_0)$ is guaranteed by Lines 4 to 6, and hence $s_0 \in \text{Cover}(C_H)$ by Lemma 4.7 when the algorithm finishes. Hence, bounding $\Delta_H$ using the definition of $H$ immediately gives the following result:

**Lemma 4.8.** *Assuming that Eq.* (6) *holds whenever* MEASURE *returns* success*, the policy $\pi$ returned by* CAPI-QPI-PLAN *is $\Delta$-optimal on $\{s_0\}$ for*

$$\Delta = 9(\varepsilon + \omega)\left(\sqrt{\tilde{d}} + 1\right)(1 - \gamma)^{-1} = \tilde{O}\left((\varepsilon + \omega)\sqrt{d}(1 - \gamma)^{-1}\right).$$

To finish the proof of Theorem 1.2, we only need to analyze the query complexity and the failure probability (i.e., the probability of Eq. (6) not being satisfied for some MEASURE call that returns *success*) of CAPI-QPI-PLAN:

*Proof of Theorem 1.2.* Both the total failure probability and query complexity of CAPI-QPI-PLAN depend on the number of times MEASURE is executed, as this is the only source of randomness and of interaction with the simulator. MEASURE can return *discover* at most $|C_0|$ times, which is bounded by $\tilde{d}$ by Eq. (13). For every $l \in [H]$, MEASURE is executed exactly once with returning *success* for each element of $C_l$. Hence, by Eq. (13) again, MEASURE returns *success* at most $\tilde{d}H$ times, each satisfying Eq. (6) with probability at least $1 - \zeta = 1 - \delta/(\tilde{d}H)$ by Lemma 4.2. By the union bound, MEASURE returns *success* in all occasions with probability at least $1 - \delta$. Hence Eq. (6) holds with probability at least $1 - \delta$, which, combined with Lemma 4.8, proves Eq. (1).

Each successful run of MEASURE executes at most $nH$ queries ($n$ is set in Line 2 of Algorithm 2). Since $H < (1 - \gamma)^{-1}\log(4\omega^{-1}(1 - \gamma)^{-1}) = \tilde{O}((1 - \gamma)^{-1})$, in total CAPI-QPI-PLAN executes at most $\tilde{O}\left(d(1 - \gamma)^{-4}\omega^{-2}\right)$ queries. As this happens at most $\tilde{d}H$ times, we obtain the desired bound on the query complexity. $\qquad\square$

## 5 Conclusions and future work

In this paper we presented CONFIDENT APPROXIMATE POLICY ITERATION, a confident version of API, which can obtain a stationary policy with a suboptimality guarantee that scales linearly with the effective horizon $H = \tilde{O}(1/(1 - \gamma))$. This scaling is optimal as shown by Scherrer and Lesner [2012].

CAPI can be applied to local planning with approximate $q^\pi$-realizability (yielding the CAPI-QPI-PLAN algorithm) to obtain a sequence of policies with successively refined accuracies on a dynamically evolving set of states, resulting in a final, recursively defined policy achieving simultaneously the optimal suboptimality guarantee and best query cost available in the literature. More precisely, CAPI-QPI-PLAN achieves $\tilde{O}(\varepsilon\sqrt{d}H)$ suboptimality, where $\varepsilon$ is the uniform policy value-function approximation error. We showed that this bound is the best (up to polylogarithmic factors) that is achievable by any planner with polynomial query cost. We also proved that the $\tilde{O}\left(dH^4\varepsilon^{-2}\right)$ query cost of CAPI-QPI-PLAN is optimal up to polylogarithmic factors in all parameters except for $H$; whether the dependence on $H$ is optimal remains an open question.

Finally, our method comes at a memory and computational cost overhead, both for the final policy and the planner. It is an interesting question if this overhead necessarily comes with the API-style method we use (as it is also present in the works of Scherrer and Lesner, 2012, Scherrer, 2014), or if it is possible to reduce it by, for example, compressing the final policy into one that is greedy with respect to some action-value function realized with the features.

## Acknowledgements

The authors would like to thank Tor Lattimore and Qinghua Liu for helpful discussions. Csaba Szepesvári gratefully acknowledges the funding from Natural Sciences and Engineering Research Council (NSERC) of Canada, "Design.R AI-assisted CPS Design" (DARPA) project and the Canada CIFAR AI Chairs Program for Amii.

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
