$$
\begin{aligned}
v^\star(s) - v^{\pi'}(s) &= v^\star(s) - q^{\pi'}(s, \pi'(s)) \\
&= v^\star(s) - q^\pi(s, \pi'(s)) + q^\pi(s, \pi'(s)) - q^{\pi'}(s, \pi'(s)) \\
&\leq v^\star(s) - q^\pi(s, \pi'(s)),
\end{aligned} \tag{14}
$$

where the first equality holds because $\pi'$ is deterministic, and the inequality is true because

$$
q^\pi(s, \pi'(s)) - q^{\pi'}(s, \pi'(s)) = \gamma \int_{s' \in \mathcal{S}} \left( v^\pi(s') - v^{\pi'}(s') \right) dP(s'|s, \pi'(s)) \leq 0
$$

by Lemma 3.1. Next observe that

$$
\hat{q}^\pi(s, \pi'(s)) \geq \max_{a \in \mathcal{A}} \hat{q}^\pi(s, a) - 2\omega \tag{15}
$$

since, as $s \notin \mathcal{S}_{\text{fix}}$, either $\pi'(s)$ is defined by Case 5a as $\pi'(s) = \arg\max_{a \in \mathcal{A}} \hat{q}^\pi(s, a)$ and so $\hat{q}^\pi(s, \pi'(s)) = \max_{a \in \mathcal{A}} \hat{q}^\pi(s, a)$, or it is defined by Case 5b in which case $\hat{q}^\pi(s, \pi'(s)) = \hat{q}^\pi(s, \pi(s)) \geq \max_{a \in \mathcal{A}} \hat{q}^\pi(s, a) - 2\omega$. Combining Eqs. (14) and (15), we obtain

$$
\begin{aligned}
v^\star(s) - v^{\pi'}(s) &\leq v^\star(s) - \hat{q}^\pi(s, \pi'(s)) + \hat{q}^\pi(s, \pi'(s)) - q^\pi(s, \pi'(s)) \\
&\leq v^\star(s) - \hat{q}^\pi(s, \pi'(s)) + \omega \\
&\leq v^\star(s) - \max_{a \in \mathcal{A}} \hat{q}^\pi(s, a) + 3\omega,
\end{aligned}
$$

where in the first line we added and subtracted $\hat{q}^\pi(s, \pi'(s))$, and the second inequality holds as $\hat{q}^\pi(s, a) \approx_\omega q^\pi(s, a)$ for $s \notin \mathcal{S}_{\text{fix}}$ and $a \in \mathcal{A}$ by the assumptions of the lemma.

We continue by adding and subtracting $\max_{a \in \mathcal{A}} q^\pi(s, a)$:

$$
\begin{aligned}
v^\star(s) - v^{\pi'}(s) &\leq v^\star(s) - \max_{a \in \mathcal{A}} q^\pi(s, a) + \max_{a \in \mathcal{A}} q^\pi(s, a) - \max_{a \in \mathcal{A}} \hat{q}^\pi(s, a) + 3\omega \\
&\leq v^\star(s) - \max_{a \in \mathcal{A}} q^\pi(s, a) + 4\omega \\
&= \max_{a \in \mathcal{A}} \left[ r(s, a) + \gamma \int_{s' \in \mathcal{S}} v^\star(s') \, dP(s'|s, a) \right] \\
&\qquad\qquad - \max_{a \in \mathcal{A}} \left[ r(s, a) + \gamma \int_{s' \in \mathcal{S}} v^\pi(s') \, dP(s'|s, a) \right] + 4\omega \\
&\leq \max_{a \in \mathcal{A}} \left[ \gamma \int_{s' \in \mathcal{S}} \left( v^\star(s') - v^\pi(s') \right) dP(s'|s, a) \right] + 4\omega \\
&\leq 4\omega + \gamma \Delta,
\end{aligned}
$$

where in the fifth line we used that $\pi$ is next-state $\Delta$-optimal by assumption. $\qquad\square$

# B Proof of Lemma 4.2

For an episode trajectory $\{S_h, A_h, R_h\}_{h \in \mathbb{N}}$, let $K$ be the smallest positive integer such that $S_K \notin \mathcal{S}'$. For any $i \in \{1, \ldots, n\}$, let $I_i$ denote the indicator of the event that at the $i^{\text{th}}$ iteration of the outer loop of Algorithm 2, the algorithm encounters $S \notin \mathcal{S}'$ in Line 6. Note that $\mathbb{E}_{\pi,s,a}[I_i] = \mathcal{P}_{\pi,s,a}[1 \leq K < H]$. Then, by Hoeffding's inequality (see, e.g., Lattimore and Szepesvári [2020]), with probability at least $1 - \zeta/2$,

$$
\left| \mathcal{P}_{\pi,s,a}[1 \leq K < H] - \frac{1}{n} \sum_{i=1}^{n} I_i \right| \leq \frac{\omega(1 - \gamma)}{4}.
$$

MEASURE only returns *success* if all indicators are zero; therefore, the above inequality implies that if MEASURE returns *success* then, with probability at least $1 - \zeta/2$, we have

$$
\mathcal{P}_{\pi,s,a}[1 \leq K < H] \leq \frac{\omega(1 - \gamma)}{4}. \tag{16}
$$

Recall that if MEASURE returns (success, $\tilde{q}$), then $\bar{q} = \frac{1}{n} \sum_{i=1}^{n} \sum_{h=0}^{H-1} \gamma^h R_{i,h}$. Since

$$0 \leq q^\pi(s,a) - \mathbb{E}_{\pi,s,a} \sum_{h=0}^{H-1} \gamma^h R_h = \mathbb{E}_{\pi,s,a} \sum_{h=H}^{\infty} \gamma^h R_h \leq \frac{\gamma^H}{1-\gamma} \leq \frac{\omega}{4},$$

another application of Hoeffding's inequality yields that $q^\pi(s,a)$ and $\bar{q}$ are close with high probability: with probability at least $1 - \zeta/2$,

$$|q^\pi(s,a) - \bar{q}| = \left| q^\pi(s,a) - \frac{1}{n} \sum_{i=1}^{n} \sum_{h=0}^{H-1} \gamma^h R_{i,h} \right|$$

$$\leq \omega/4 + \left| \mathbb{E}_{\pi,s,a} \sum_{h=0}^{H-1} \gamma^h R_h - \frac{1}{n} \sum_{i=1}^{n} \sum_{h=0}^{H-1} \gamma^h R_{i,h} \right| \leq \omega/2, \tag{17}$$

where we also used that the range of the sum of the rewards above for every $i$ is $[0, 1/(1-\gamma)]$.

Pick any $\pi' \in \Pi_{\pi,S'}$. Observe that for any $s \in S$ and $a \in \mathcal{A}$, the distribution of the trajectory $S_0, A_0, R_0, S_1, A_1, R_1, \ldots, A_{K-1}, R_{K-1}, S_K$ is the same under $\mathcal{P}_{\pi',s,a}$ and $\mathcal{P}_{\pi,s,a}$, as $\pi$ and $\pi'$ select the same actions for states in $S'$. By Eqs. (3) to (4), we can write

$$\left| q^{\pi'}(s,a) - q^\pi(s,a) \right| = \left| \mathbb{E}_{\pi',s,a} \left[ \sum_{t \in [K]} \gamma^t R_t + \gamma^K v^{\pi'}(S_K) \right] - \mathbb{E}_{\pi,s,a} \left[ \sum_{t \in [K]} \gamma^t R_t + \gamma^K v^{\pi'}(S_K) \right] \right|$$

$$= \left| \mathbb{E}_{\pi,s,a} \left[ \gamma^K \left( v^{\pi'}(S_K) - v^\pi(S_K) \right) \right] \right| \leq \frac{1}{1-\gamma} \mathbb{E}_{\pi,s,a} \left[ \gamma^K \right]$$

$$\leq \frac{1}{1-\gamma} \mathcal{P}_{\pi,s,a} \left[ 1 \leq K < H \right] + \frac{\gamma^H}{1-\gamma} \leq \frac{1}{1-\gamma} \mathcal{P}_{\pi,s,a} \left[ 1 \leq K < H \right] + \omega/4. \tag{18}$$

Combining Eqs. (16) to (18), it follows by the union bound that if MEASURE returns with (success, $\tilde{q}$), then with probability at least $1 - \zeta$,

$$\left| q^{\pi'}(s,a) - \bar{q} \right| \leq \left| q^{\pi'}(s,a) - q^\pi(s,a) \right| + |q^\pi(s,a) - \bar{q}| \leq \omega. \qquad \square$$

## C Proof of Lemma 4.3

We start the proof by showing that there exists a $\theta \in \mathbb{R}^d$ such that

$$\|\theta\|_2 \leq B \text{ and for all } s \in S \text{ and } a \in \mathcal{A}, q^\pi(s,a) \approx_\varepsilon \langle \theta, \varphi(s,a) \rangle. \tag{19}$$

For any finite set $W \subseteq S \times \mathcal{A}$, $\max_{(s,a) \in W} |q^\pi(s,a) - \langle \varphi(s,a), \theta' \rangle|$ is a continuous function of $\theta'$, hence it attains its infimum on the compact set $\{\theta' \in \mathbb{R}^d : \|\theta'\|_2 \leq B\}$. By Definition 1.1, this infimum is at most $\varepsilon$. Therefore, the compact sets $\Theta_{s,a} = \{\theta' \in \mathbb{R}^d : \|\theta'\|_2 \leq B \text{ and } |q^\pi(s,a) - \langle \varphi(s,a), \theta' \rangle| \leq \varepsilon\}$ are non-empty for all $(s,a) \in S \times \mathcal{A}$, and any intersection of a finite collection of these sets is also non-empty. Therefore, $\bigcap_{(s,a) \in S \times \mathcal{A}} \Theta_{s,a}$ is non-empty by [Rudin et al., 1976, Theorem 2.36], and any element $\theta$ of this set satisfies Eq. (19). For the remainder of this proof, fix such a $\theta$.

For any $i \in [n]$, with a slight abuse of notation, we introduce the shorthand $\varphi_i = \varphi(s_i, a_i)$, and let $\hat{q}_i = \langle \theta, \varphi_i \rangle$ and $\xi_i = \bar{q}_i - \hat{q}_i$. Note that by the triangle inequality, $|\xi_i| \leq |\bar{q}_i - q^\pi(s_i, a_i)| + |q^\pi(s_i, a_i) - \hat{q}_i| \leq \omega + \varepsilon$. Let $\bar{\theta} = V(C)^{-1} \sum_{i \in [n]} \varphi_i \bar{q}_i$ and $\hat{\theta} = V(C)^{-1} \sum_{i \in [n]} \varphi_i \hat{q}_i$.

For any $v \in \mathbb{R}^d$ by the Cauchy-Schwarz inequality,

$$\left| \langle \bar{\theta} - \theta, v \rangle \right| \leq \left| \langle \hat{\theta} - \theta, v \rangle \right| + \left| \langle \bar{\theta} - \hat{\theta}, v \rangle \right| \leq \|v\|_{V(C)^{-1}} \|\hat{\theta} - \theta\|_{V(C)} + \left| \left\langle V(C)^{-1} \sum_{i \in [n]} \varphi_i \xi_i, v \right\rangle \right|.$$

To bound the first term on the right-hand side above, observe that

$$\|\hat{\theta} - \theta\|_{V(C)} = \left\| V(C)^{-1} \left( \sum_{i \in [n]} \varphi_i \varphi_i^\top \right) \theta - \theta \right\|_{V(C)} = \lambda \|\theta\|_{V(C)^{-1}} \leq \lambda \|\theta\|_{\frac{1}{\lambda} \mathbb{I}} \leq \sqrt{\lambda} B,$$

where in the last line we used that $V(C) \succeq \lambda \mathbb{I}$.

The second term can be bounded as

$$\left| \left\langle V(C)^{-1} \sum_{i \in [n]} \varphi_i \xi_i, v \right\rangle \right| \leq \sum_{i \in [n]} \left| \langle V(C)^{-1} \varphi_i \xi_i, v \rangle \right|$$

$$\leq (\omega + \varepsilon) \sum_{i \in [n]} \left| \langle V(C)^{-1} \varphi_i, v \rangle \right|$$

$$\leq (\omega + \varepsilon) \sqrt{n} \sqrt{\sum_{i \in [n]} \left( \langle V(C)^{-1} \varphi_i, v \rangle \right)^2}$$

$$\leq (\omega + \varepsilon) \sqrt{n} \sqrt{v^\top V(C)^{-1} \left( \sum_{i \in [n]} \varphi_i \varphi_i^\top \right) V(C)^{-1} v + v^\top V(C)^{-1} \lambda \mathbb{I} V(C)^{-1} v}$$

$$= (\omega + \varepsilon) \sqrt{n} \, \|v\|_{V(C)^{-1}} \,,$$

where the first inequality holds by the triangle inequality, the second by our bound on $|\xi_i|$, the third by the Cauchy-Schwartz inequality, and the fourth by the positivity of $\lambda$. Putting it all together, for any $s \in \mathcal{S}$ and $a \in \mathcal{A}$, using the previous bounds with $v = \varphi(s, a)$,

$$\left| \text{LSE}_{C, \bar{q}}(s, a) - q^\pi(s, a) \right| \leq |q^\pi(s, a) - \langle \theta, \varphi(s, a) \rangle| + \left| \langle \bar{\theta} - \theta, \varphi(s, a) \rangle \right|$$

$$\leq \varepsilon + \|\varphi(s, a)\|_{V(C)^{-1}} \left( \sqrt{\lambda} B + (\omega + \varepsilon) \sqrt{n} \right) \,,$$

completing the proof. $\qquad \square$

## D   Deriving next-state optimality of $\pi_\ell$ for Lemma 4.4

**Lemma D.1.** *Assume that Eq. (6) holds whenever* MEASURE *returns success. At any point of* CAPI-QPI-PLAN *after Line 16 is executed, for any $\pi'' \in \Pi_{\pi_\ell, \text{Cover}(C_\ell)}$, $s \in \text{Cover}(C_\ell)$, and $a \in \mathcal{A}$,*

$$\left| \hat{q}(s, a) - q^{\pi''}(s, a) \right| \leq (\omega + \varepsilon)(\sqrt{\tilde{d}} + 1) \,.$$

*Proof.* By Lemma 4.6 and Eq. (6), $\bar{q}_{l,m} \approx_\omega q^{\pi''}(C_{l,m})$ for all $m \in [|C_\ell|]$ (recall that $C_{l,m}$ is the $m^{\text{th}}$ state-action pair in $C_l$). Therefore, applying Lemma 4.3 with $q^{\pi''}$, $C_\ell$ and $\bar{q}_\ell$, as $\hat{q} = \text{LSE}_{C_\ell, \bar{q}_\ell}$, we get that for any $s \in \text{Cover}(C_\ell)$ and all $a \in \mathcal{A}$,

$$\left| \hat{q}(s, a) - q^{\pi''}(s, a) \right| \leq \varepsilon + \|\varphi(s, a)\|_{V(C_\ell)^{-1}} \left( \sqrt{\lambda} B + (\omega + \varepsilon) \sqrt{|C_\ell|} \right)$$

$$\leq (\omega + \varepsilon)(\sqrt{\tilde{d}} + 1) \,,$$

where the second inequality holds because $\|\varphi(s, a)\|_{V(C_\ell)^{-1}} \leq 1$ since $s \in \text{Cover}(C_\ell)$, $|C_\ell| \leq \tilde{d}$ by Eq. (13), and the definition of $\lambda$. $\qquad \square$

**Lemma D.2.** *Assume that Eq. (6) holds whenever* MEASURE *returns success. Consider a time when Lines 17 to 20 of* CAPI-QPI-PLAN *are run and assume that at this time, for all $l \in [H + 1]$, $\pi_l$ is $\Delta_l$-optimal on $\text{Cover}(C_l)$. Then, $\pi_\ell$ is next-state $(\Delta_\ell + 4(\omega + \varepsilon)(\sqrt{\tilde{d}} + 1)/\gamma)$-optimal on $\text{Cover}(C_\ell)$.*

*Proof.* Let $\pi_\ell^+$ be defined as in Eq. (22). As $\pi_\ell^+ \in \Pi_{\pi_\ell, \text{Cover}(C_\ell)}$, by Lemma D.1, for any $s \in \text{Cover}(C_\ell)$ and all $a \in \mathcal{A}$,

$$\left| \hat{q}(s, a) - q^{\pi_\ell^+}(s, a) \right| \leq (\omega + \varepsilon)(\sqrt{\tilde{d}} + 1) \,.$$

Similarly, applying Lemma D.1 with $\pi_\ell$ (which trivially belongs to $\Pi_{\pi_\ell, \text{Cover}(C_\ell)}$), we also have

$$|\hat{q}(s, a) - q^{\pi_\ell}(s, a)| \leq (\omega + \varepsilon)(\sqrt{\tilde{d}} + 1) \,.$$

Therefore,

$$\left| q^{\pi_\ell^+}(s,a) - q^{\pi_\ell}(s,a) \right| \le 2(\omega + \varepsilon)(\sqrt{\bar{d}} + 1). \tag{20}$$

Since $\pi_\ell$ is $\Delta_\ell$-optimal on $\mathrm{Cover}(C_\ell)$ by assumption, this makes $\pi_\ell^+$ $\Delta$-optimal on $\mathrm{Cover}(C_\ell)$ for

$$\Delta = \Delta_\ell + 2(\omega + \varepsilon)(\sqrt{\bar{d}} + 1). \tag{21}$$

For a trajectory in the MDP, let the random variable $\tau$ be the first time the state is in $\mathrm{Cover}(C_\ell)$:

$$\tau = \min\{t \in \mathbb{N} \mid S_t \in \mathrm{Cover}(C_\ell)\}.$$

Since $\pi_\ell^+$ agrees with $\pi^\star$ on states not in $\mathrm{Cover}(C_\ell)$, the distribution of the trajectory up to and including $S_\tau$ is the same under both policies, starting from any state $s \in \mathcal{S}$. Therefore, for any $s \in \mathcal{S}$,

$$v^\star(s) - v^{\pi_\ell^+}(s) = \mathbb{E}_{\pi^\star,s}\left[\sum_{t \in \mathbb{N}} \gamma^t R_t\right] - \mathbb{E}_{\pi_\ell^+,s}\left[\sum_{t \in \mathbb{N}} \gamma^t R_t\right]$$

$$= \mathbb{E}_{\pi_\ell^+,s}\left[\gamma^\tau \left(v^\star(S_\tau) - v^{\pi_\ell^+}(S_\tau)\right)\right]$$

$$\le \Delta,$$

as $\gamma^\tau \le 1$ and $\pi_\ell^+$ is $\Delta$-optimal on $\mathrm{Cover}(C_\ell)$. That is, $\pi_\ell^+$ is also $\Delta$-optimal on $\mathcal{S}$ (with $\Delta$ defined in Eq. 21). Using this, for any $s \in \mathrm{Cover}(C_\ell)$, and $a \in \mathcal{A}$, we have

$$\int_{s' \in \mathcal{S}} \left(v^\star(s') - v^{\pi_\ell}(s')\right) \mathrm{d}P(s'|s,a)$$

$$\le \int_{s' \in \mathcal{S}} \left(v^\star(s') - v^{\pi_\ell^+}(s')\right) \mathrm{d}P(s'|s,a) + \left|\int_{s' \in \mathcal{S}} \left(v^{\pi_\ell^+}(s') - v^{\pi_\ell}(s')\right) \mathrm{d}P(s'|s,a)\right|$$

$$\le \Delta_\ell + 2(\omega + \varepsilon)(\sqrt{\bar{d}} + 1) + \frac{1}{\gamma}\left|q^{\pi_\ell^+}(s,a) - q^{\pi_\ell}(s,a)\right|$$

$$\le \Delta_\ell + 2(\omega + \varepsilon)(\sqrt{\bar{d}} + 1) + 2(\omega + \varepsilon)(\sqrt{\bar{d}} + 1)/\gamma$$

$$= \Delta_\ell + 4(\omega + \varepsilon)(\sqrt{\bar{d}} + 1)/\gamma,$$

where the third inequality holds by Eq. (20). Therefore $\pi_\ell$ is next-state $(\Delta_\ell + 4(\omega + \varepsilon)(\sqrt{\bar{d}} + 1)/\gamma)$-optimal on $\mathrm{Cover}(C_\ell)$. $\qquad\square$

## E   Poof of Lemma 4.4

*Proof of Lemma 4.4.* We prove by induction on the iterations of the main loop of CAPI-QPI-PLAN the *inductive hypothesis:* at the start of iteration $i$, for all $l \in [H + 1]$, $\pi_l$ is $\Delta_l$-optimal on $\mathrm{Cover}(C_l)$. We first observe that after initialization, $C_l$ is the empty sequence for every $l$, so we can apply Lemma 4.3 with $q^\star$ and empty sequences ($n = 0$) to get that for any $s \in \mathrm{Cover}(())$ and $a \in \mathcal{A}$, $q^\star(s,a) \le \varepsilon + \sqrt{\lambda}B = \varepsilon + \omega$. Then, $v^\star(s) \le \varepsilon + \omega \le \Delta_l$. Therefore, at initialization, any policy is $\Delta_l$-optimal on $\mathrm{Cover}(C_l)$ for any $l \in [H + 1]$.

Assuming that the inductive hypothesis holds at the start of some iteration, it is left to prove that it continues to hold at the end of the iteration (assuming Eq. (6) holds whenever MEASURE returns *success*); this implies that the hypothesis also holds at the start of the next iteration and hence also proves the lemma. For any $(s, a)$ appended to $C_0$, the inductive hypothesis trivially continues to hold as $\Delta_0 = 1/(1 - \gamma) \ge v^\star(s)$ for any $s \in \mathcal{S}$ because the rewards are bounded in $[0, 1]$. The only other case in which $C_l$ or $\pi_l$ changes for any $l$ is in Lines 18 and 20, where the changes happen only for $l = \ell + 1$.

We will use Lemma 3.4 to analyze the effect of these updates, thus next we show that the conditions of the lemma are satisfied:

*(a)* In Lemma D.2 we show that $\pi_\ell$ is next-state $(\Delta_\ell + 4(\omega + \varepsilon)(\sqrt{\bar{d}} + 1)/\gamma)$-optimal on $\mathrm{Cover}(C_\ell)$. In the proof of the lemma, we introduce a policy in Eq. (22) that acts as $\pi_\ell$ on states in $\mathrm{Cover}(C_\ell)$, and as an optimal stationary deterministic memoryless policy $\pi^\star$ otherwise:

$$\pi_\ell^+(s) = \begin{cases} \pi_\ell(s) & \text{if } s \in \mathrm{Cover}(C_\ell); \\ \pi^\star(s) & \text{otherwise}. \end{cases} \tag{22}$$

Intuitively, this policy corrects $\pi_\ell$ on the low-confidence states. The proof of Lemma D.2 then uses the fact that this policy is also $q^\pi$-realizable (Definition 1.1) and satisfies $\pi_\ell^+ \in \Pi_{\pi_\ell,\text{Cover}(C_\ell)}$ to show (i) that the $q$-values of $\pi_\ell$ and $\pi_\ell^+$ are close on the measured state-action pairs (via Lemma 4.6 and Lemma D.1); (ii) an optimality guarantee on $\pi_\ell^+$ for all $s \in \mathcal{S}$; and, as a consequence, (iii) the next-state optimality of $\pi_\ell$.

*(b)* Next, to analyze the effect of Line 18, we introduce hypothetical $q$-approximators $\tilde{q}_l$ for $l \in [H+1]$, defined as follows: At initialization, $\tilde{q}_l(s, a) = 0$ for all $l \in [H+1]$, $s \in \mathcal{S}$, and $a \in \mathcal{A}$. It is updated every time after Line 16 of the algorithm is executed as

$$\tilde{q}_\ell(s, a) \leftarrow \begin{cases} \tilde{q}_\ell(s, a) & \text{if } s \in \text{Cover}(C_{\ell+1}); & \text{(23a)} \\ \hat{q}(s, a) & \text{otherwise.} & \text{(23b)} \end{cases}$$

In other words, $\tilde{q}_\ell$ is only updated to the newly computed $\hat{q}$ for states that are not in $\text{Cover}(C_{\ell+1})$, and stays unchanged for other states. We show in Lemma F.2 that the new policy that $\pi_{\ell+1}$ is updated to, which is constructed in two steps (Lines 17–18), can be expressed as the result of a *single* CAPI policy update that uses $\tilde{q}$:

$$\pi_{\ell+1} \leftarrow \pi_{\tilde{q}_\ell, \pi_\ell, \mathcal{S}\setminus\text{Cover}(C_l)} \,.$$

We show in Lemma F.1 that $\tilde{q}_\ell \approx_{\omega'} q^{\pi_\ell}$ with $\omega' = (\omega + \varepsilon)(\sqrt{\bar{d}} + 1)$ on $\text{Cover}(C_\ell)$.

By the above, we can apply Lemma 3.4 with policy $\pi_\ell$, $q$-approximation $\tilde{q}_\ell$ (with approximation error guarantee $\omega'$ on $\text{Cover}(C_\ell)$, and $\mathcal{S}_{\text{fix}} = \mathcal{S} \setminus \text{Cover}(C_\ell)$ to get that the new value of $\pi_{\ell+1}$ is a $\Delta_{\ell+1} = (8(\omega + \varepsilon)(\sqrt{\bar{d}} + 1) + \gamma\Delta_\ell)$-optimal policy on $\text{Cover}(C_\ell)$. By the end of the loop in Line 20, $\text{Cover}(C_{\ell+1}) = \text{Cover}(C_\ell)$, so $\pi_{\ell+1}$ is $\Delta_{\ell+1}$-optimal on $\text{Cover}(C_{\ell+1})$. This finishes the proof that the inductive hypothesis continues to hold at the end of the iteration, finishing the proof of the lemma. □

# F   Auxiliary results for Lemma 4.4 about $\tilde{q}_l$

Throughout the execution of CAPI-QPI-PLAN, for $l \in [H+1]$, let $\tilde{q}_l^-, \pi_l^-, C_l^-$ denote the values of variables $\tilde{q}_\ell, \pi_\ell, C_\ell$, respectively, at the time when Lines 16–20 were most recently executed with $\ell = l$ in a previous iteration of the main loop of CAPI-QPI-PLAN. If such a time does not exist, let their values be the initialization values. Thus, $C_l^-$ may (only) change at the start of some iteration $i$ if Lines 16–20 were executed with $\ell = l$ in the previous iteration $i - 1$. Observe that whenever this happens, Lines 16–20 may also change $C_{\ell+1}$ in iteration $i - 1$, and this is the only time $C_{l+1}$ can be changed for any $l \in [H]$. After this, at the beginning of iteration $i$, $C_{l+1}$ always has the same elements as $C_l^-$. Therefore, since it also holds at the initialization of the algorithm, we conclude that at the start of each iteration,

$$\text{Cover}(C_{l+1}) = \text{Cover}(C_l^-) \,. \tag{24}$$

**Lemma F.1.** *Assume that Eq. (6) holds whenever* MEASURE *returns* success. *Then, whenever Line 18 of* CAPI-QPI-PLAN *is executed, for all $s \in \text{Cover}(C_\ell)$ and $a \in \mathcal{A}$,*

$$\left| \tilde{q}_\ell(s, a) - q^{\pi''}(s, a) \right| \le (\omega + \varepsilon)(\sqrt{\bar{d}} + 1) \qquad \text{for all } \pi'' \in \Pi_{\pi_\ell, \text{Cover}(C_\ell)} \,. \tag{25}$$

*Proof.* We prove this by induction for every time Line 18 is executed with any value of $\ell$. We first observe that after initialization, $C_l$ is the empty sequence for every $l$, so we can apply Lemma 4.3 with $q^\star$ and empty sequences ($n = 0$) to get that for any $s \in \text{Cover}(())$ and $a \in \mathcal{A}$, $q^{\pi''}(s, a) \le q^\star(s, a) \le \varepsilon + \sqrt{\lambda}B = \varepsilon + \omega$. Also, $\tilde{q}_l(\cdot, \cdot) = 0$ at initialization, so Eq. (25) holds for any value of $\ell$.

Consider a time when Line 18 is executed and assume the inductive hypothesis holds for the previous time Line 18 was executed with the same value of $\ell$ (or at the initialization if this is the first time), that is,

$$\left| \tilde{q}_\ell^-(s, a) - q^{\pi''}(s, a) \right| \le (\omega + \varepsilon)(\sqrt{\bar{d}} + 1) \qquad \text{for all } \pi'' \in \Pi_{\pi_\ell^-, \text{Cover}(C_\ell^-)}, \ s \in \text{Cover}(C_\ell^-) \,.$$

To prove that the statement now holds for any $s \in \text{Cover}(C_\ell)$, first consider any $s \in \text{Cover}(C_{\ell+1}) = \text{Cover}(C_\ell^-)$. For such an $s$, by Lemma 4.5 we have that $\Pi_{\pi_\ell, \text{Cover}(C_\ell)} \subseteq \Pi_{\pi_\ell^-, \text{Cover}(C_\ell^-)}$. Also, by definition, $\tilde{q}_\ell(s, \cdot) = \tilde{q}_\ell^-(s, \cdot)$ for $s \in \text{Cover}(C_{\ell+1})$. Combining with the inductive hypothesis, it follows that Eq. (25) holds for $s \in \text{Cover}(C_{\ell+1})$.

It remains to show that Eq. (25) also holds for $s \in \text{Cover}(C_\ell) \setminus \text{Cover}(C_{\ell+1})$. For such an $s$, $\tilde{q}_\ell(s, \cdot) = \hat{q}(s, \cdot)$ by definition, and hence Lemma D.1 implies that Eq. (25) holds in this case.

Combining the two cases, it follows that the inductive hypothesis continues to hold when Line 18 is executed. $\qquad\square$

**Lemma F.2.** *Throughout the execution of* CAPI-QPI-PLAN*, at the start of any iteration, for all* $l \in [H]$,

$$\pi_{l+1} = \pi_{\tilde{q}_l^-, \pi_l^-, \mathcal{S} \setminus \text{Cover}(C_l^-)} . \tag{26}$$

*Proof.* We prove this by induction for the start of any iteration. Eq. (26) holds at the start of the algorithm due to its initialization (because at initialiaztion, $\tilde{q}_l^-(s, a) = 0$ for all $s, a$, and hence by our tie-breaking rule, the policy on the right-hand side of Eq. (26) always chooses action $\mathcal{A}_1$, which is the initial policy for $\pi_l$).

In what follows, we use the fact that for any $q : \mathcal{S} \times \mathcal{A} \to \mathbb{R}$, policy $\pi$, and $\mathcal{S}_{\text{fix}} \subseteq \mathcal{S}$, the CAPI policy update $\pi_{q, \pi, \mathcal{S}_{\text{fix}}}$ is a policy whose value at any $s \in \mathcal{S}$ only depends on $q(s, \cdot)$, $\pi(s)$, and whether or not $s \in \mathcal{S}_{\text{fix}}$, by definition (Eq. 5). Therefore, for an alternative $q'$, $\pi'$, $\mathcal{S}'_{\text{fix}}$, for any $s \in \mathcal{S}$, $\pi_{q, \pi, \mathcal{S}_{\text{fix}}}(s) = \pi_{q', \pi', \mathcal{S}'_{\text{fix}}}(s)$ whenever the following three conditions hold: (C1) $q(s, a) = q'(s, a)$ for all $a \in \mathcal{A}$; (C2) $\pi(s) = \pi'(s)$; and (C3) either both or none of $\mathcal{S}_{\text{fix}}$ and $\mathcal{S}'_{\text{fix}}$ include $s$.

Assume the inductive hypothesis holds at the beginning of some iteration. Let $\pi''$ be the policy Line 18 updates $\pi_{\ell+1}$ to, noting that this is the only place where policies are updated. All we need to prove is that $\pi''$ is equal to

$$\tilde{\pi} = \pi_{\tilde{q}_\ell, \pi_\ell, \mathcal{S} \setminus \text{Cover}(C_\ell)} .$$

First, for any $s \notin \text{Cover}(C_{\ell+1})$, $\pi''(s) = \pi'(s) = \pi_{\hat{q}, \pi_\ell, \mathcal{S} \setminus \text{Cover}(C_\ell)}(s)$ and $\hat{q}(s, \cdot) = \tilde{q}_\ell(s, \cdot)$ by definition. Hence, $\pi''(s) = \pi_{\hat{q}, \pi_\ell, \mathcal{S} \setminus \text{Cover}(C_\ell)}(s) = \pi_{\tilde{q}_\ell, \pi_\ell, \mathcal{S} \setminus \text{Cover}(C_\ell)}(s) = \tilde{\pi}(s)$, as all of conditions (C1)-(C3) are satisfied for $s$ (C2 and C3 hold trivially).

Next, take any $s \in \text{Cover}(C_{\ell+1}) = \text{Cover}(C_\ell^-)$. Then, by Line 18, $\pi''(s) = \pi_{\ell+1}(s)$. By the inductive hypothesis, the current value of $\pi_{\ell+1}$ can be written as $\pi_{\tilde{q}_\ell^-, \pi_\ell^-, \mathcal{S} \setminus \text{Cover}(C_\ell^-)}$. We prove that this policy takes the same value as $\tilde{\pi}$ at $s$, by showing conditions (C1)-(C3). First, by Lemma 4.5, $\pi_\ell \in \Pi_{\pi_\ell^-, \text{Cover}(C_\ell^-)}$. Thus, as $s \in \text{Cover}(C_\ell^-)$, $\pi_\ell(s) = \pi_\ell^-(s)$, showing condition (C2). Furthermore, as $s \in \text{Cover}(C_{\ell+1})$, by definition, $\tilde{q}_\ell(s, \cdot) = \tilde{q}_\ell^-(s, \cdot)$, showing condition (C1). Finally, as $s \in \text{Cover}(C_{\ell+1}) = \text{Cover}(C_\ell^-) \subseteq \text{Cover}(C_\ell)$, $s \notin \mathcal{S} \setminus \text{Cover}(C_\ell^-)$ and $s \notin \mathcal{S} \setminus \text{Cover}(C_\ell)$, showing condition (C3).

Combining the two cases, $\pi''(s) = \tilde{\pi}(s)$ for any $s \in \mathcal{S}$, finishing the induction. $\qquad\square$

# G   Efficient implementation and proof of Theorem 1.3

In this section we consider the efficient implementation of CAPI-QPI-PLAN in terms of memory and computational costs of both the algorithm itself and the final policy it outputs.

Focusing on the memory cost, first we can observe that throughout the execution of the algorithm, $C_l$ for all $l \in [H + 1]$ only stores up to $\tilde{d}$ unique state-action pairs altogether (cf. Eq. (13)), as they use the same pairs; let $W = (s_i, a_i)_{i \in \hat{d}}$ denote these for some $\hat{d} \le \tilde{d}$. Furthermore, throughout the execution of the algorithm, for any level $l$, the only features that $\pi_l$ depends on are the features associated with members of $W$. Storing all these features takes $d\hat{d}$ memory. Denote all the policies that CAPI-QPI-PLAN constructs in Line 18, in order, as $\pi^{(0)}, \pi^{(1)}, \ldots, \pi^{(n-1)}$, where $n$ is the number of times Line 18 is executed. Recall from the proof of Theorem 1.2 that the number of times MEASURE returns *success*, which is an upper bounds on $n$, is itself bounded by $\tilde{d}H$, hence $n \le \tilde{d}H$. Together, Lines 17-18 construct a policy that, for an $s \in \mathcal{S}$, decides whether the action should be $\arg\max_{a \in \mathcal{A}} \langle \varphi(s, a), \theta \rangle$ for some $\theta$ given by LSE (Eq. (8)), or the value of the policy should be determined by a recursive call to a previously constructed policy, either $\pi_{\ell+1}$ or $\pi_\ell$ (through $\pi'$). Now there exist some $a, b \in [n]$ such that $\pi^{(a)} = \pi_\ell$ and $\pi^{(b)} = \pi_{\ell+1}$ before the new policy is constructed in Line 18. To implement the new $\pi_{\ell+1}$ constructed policy, it is enough therefore to store, in addition to the existing policies, $\theta$ (from $\hat{q}$), the decision rules, and the indices $a$ and $b$. The decision rules are fully defined by $\theta$, $C_\ell$, and $C_{\ell+1}$. It is therefore enough to further store $C_\ell, C_{\ell+1} \subseteq W$, which can be

encoded as $\hat{d}$-dimensional vectors each, storing the bitmask of which state-action pairs are included. We also store the current value of $\ell$ (the level) for the newly constructed policy. Together, a policy thus consumes $3 + d + 2\hat{d}$ memory. We store all policies constructed, along with the features of $W$, and the final value of $V(C_H)^{-1}$, at a memory cost of $d\hat{d} + \tilde{d}H(3 + d + \hat{d}) + d^2 = \tilde{O}(d^2/(1 - \gamma))$. This is the memory cost of the final policy outputted by CAPI-QPI-PLAN. The memory cost of running CAPI-QPI-PLAN itself is of the same order, as additionally storing $C_l$, $\bar{q}_l$, and $V(C_l)^{-1}$ for $l \in [H + 1]$ takes $\tilde{O}(d^2/(1 - \gamma))$ memory.

To efficiently implement the final policy found by CAPI-QPI-PLAN with the stored information described above, we start from evaluating the last policy constructed, $\pi^{(i)}$ for $i = n - 1$. We introduce auxiliary variables $\tilde{V}(C_l)^{-1}$ and $\tilde{C}_l$ for $l \in [H + 1]$ to efficiently track the required values of $V(C_l)^{-1}$ and $C_l$. We keep updating these variables so that for $l \in \{\ell, \ell + 1\}$, they match the values of $V(C_l)^{-1}$ and $C_l$, respectively, at the time of construction of the current policy $\pi^{(i)}$ under consideration, where $\ell$ is the (saved) level of $\pi^{(i)}$. For $i = n - 1$, observe that when it was constructed, $C_0 = C_1 = \cdots = C_H$ by Lemma 4.7. We therefore start by initializing variables $\tilde{V}(C_0)^{-1}, \ldots, \tilde{V}(C_H)^{-1}$ to the saved final value of $V(C_H)^{-1}$, and variables $\tilde{C}_0, \ldots, \tilde{C}_H$ to $W$. Implementing the decisions of a policy takes an order of $|\mathcal{A}|d^2$ computation ($|\mathcal{A}|$ vector and matrix multiplications), after which we recover either the policy output or a previously constructed policy to recurse into. For the latter case, we have to consider the evaluation of this policy, denoted by $\pi^{(i')}$. Let the (saved) level of $\pi^{(i')}$ be $\ell'$. Before we set $i$ to $i'$ and start evaluating it, we need to update the values of $\tilde{V}(C_l)$ and $C_l$ for $l \in \{\ell', \ell' + 1\}$. The updates are needed for these two levels only, as the decision rule of policy $i'$ only depends on these levels, as shown before. Let us describe the update procedure for some $l \in \{\ell', \ell' + 1\}$: Since $\pi^{(i')}$ was constructed earlier than $\pi^{(i)}$ (i.e., $i' < i$), and $C_{l'}$ can only grow during the algorithm for any $l' \in [H + 1]$, we only need to remove members of the variable $\tilde{C}_l$ to match the value of $C_l$ at the time of construction of $\pi^{(i')}$. The members to be removed are given by the difference of the members of $\tilde{C}_l$ and the bitmasks stored for $\pi^{(i')}$ for level $l$. For each state-action pair $(s, a)$ removed, we also need to update $\tilde{V}(C_l)^{-1}$ to $\left(\tilde{V}(C_l) - \varphi(s, a)\varphi(s, a)^\top\right)^{-1}$, which can be done in order $d^2$ computation using the Sherman–Morrison–Woodbury formula [Max, 1950]. The total number of such removal operations for any level $l$ is bounded by the sum of the number of state-action pairs in the initialization of $\tilde{C}_{l'}$ (for $l' \in [H + 1]$), that is, by $(H + 1)\hat{d}$. As a result, the computational cost of the final policy of CAPI-QPI-PLAN is $\tilde{O}((H + 1)\hat{d}d^2) + n\tilde{O}(|\mathcal{A}|d^2) = \tilde{O}(d^3|\mathcal{A}|/(1 - \gamma))$.

Finally, we consider the computational cost of running CAPI-QPI-PLAN. The number of iterations of the outer loop is bounded by $\tilde{O}(dH) = \tilde{O}(d/(1 - \gamma))$, as each iteration involves either a MEASURE call that returns *success*, or a new member added to some $C_l$. For each iteration, Line 4 takes $\tilde{O}(d^2|\mathcal{A}|)$, Line 7 takes $\tilde{O}(d/(1 - \gamma))$, Line 11 takes $\tilde{O}(d^2|\mathcal{A}|)$ computation; for Line 16, calculating $\theta$, the second component of the inner product of the least-squares predictor in Eq. (8) takes $\tilde{O}(d^2)$ computation, and if $C_l$ ever changes for some $l$, updating $V(C_l)^{-1}$ by the Sherman–Morrison–Woodbury takes $\tilde{O}(d^2)$ computation. Overall, all the operations except those associated to the MEASURE call of Line 9 take $\tilde{O}(d^3|\mathcal{A}|/(1 - \gamma))$ computation in total. We conclude our calculations by considering the computational cost of the MEASURE calls, which will dominate the overall computational cost. Line 6 of Algorithm 2 has a computational cost of order $d^2|\mathcal{A}|$, while the majority of the computational cost comes from evaluating the policy at Line 7. By our previous calculations, this takes $\tilde{O}(d^3|\mathcal{A}|/(1 - \gamma))$ computation and happens (at most) once for each simulator call. Using the query cost bound of Theorem 1.2, we conclude that the computational cost of CAPI-QPI-PLAN is $\tilde{O}(d^4|\mathcal{A}|(1 - \gamma)^{-5}\omega^{-2})$. $\qquad\square$

## H   Query cost lower bounds with random access

In this section we prove lower bounds on the worst-case expected query cost of planning algorithms with a simulator supporting *random access*. Recall from Section 1 that in this setting a planner can issue queries for any state-action pair, not just the ones already visited. As this is a more powerful access to the simulator than *local access*, statements that hold for *all* planners using *random access* (as such, all lower bounds presented in this section) trivially hold for planners using *local access*. We prove two bounds, Theorem H.2 and Theorem H.3, whose combination trivially implies Theorem 1.4.

Formally, the planner interacts with a *random access* simulator that simulates some MDP $M$ as follows: at step $t$ starting from 1, given the whole interaction history $H_t =$

$(S_1, A_1, R_1, S'_1, \ldots, S_{t-1}, A_{t-1}, R_{t-1}, S'_{t-1})$ (where $H_1$ is the empty sequence by definition), the planner either selects a state-action pair $(S_t, A_t)$, or halts and outputs a stationary memoryless policy. The planner is allowed to randomize. Let $\tau$ denote the number of queries the planner sends to the simulator before it halts, and $\pi_\tau$ the policy it outputs. If the planner does not stop, the simulator responds to the query $(S_t, A_t)$ by returning $(S'_t, R_t)$ sampled independently from the transition-reward kernel $Q(S_t, A_t)$ of $M$. Let $\mathcal{P}_M$ denote the probability measure associated with this procedure, and let $\mathbb{E}_M$ denote the expectation operator corresponding to $\mathcal{P}_M$. Both $\mathcal{P}_M$ and $\mathbb{E}_M$ implicitly depend on the planner, which is omitted in the notation for brevity but will always be clear from the context. Using this notation, clearly $\mathbb{E}_M(\tau)$ is the expected query cost of the planner on $M$.

As usual, we only consider the query complexity of planners which are reasonable in the sense that they can find a near-optimal policies for a class of MDPs:

**Definition H.1** (Soundness and query complexity). *A planner is said to be $(\alpha, \delta)$-sound for an MDP $M$ if, when used with a simulator of $M$, it halts almost surely (i.e., $\mathcal{P}_M(\tau < \infty) = 1$) and outputs a policy $\pi_\tau$ that is $\alpha$-optimal for $M$ with probability at least $1 - \delta$, that is,*

$$\mathcal{P}_M\left(v^\star(s_0) - v^{\pi_\tau}(s_0) \le \alpha\right) \ge 1 - \delta,$$

*where $v^\star$ and $v^{\pi_\tau}$ are the value-functions of the optimal policy and $\pi_\tau$ in the MDP $M$ and $s_0$ is the initial state of $M$. A planner is $(\alpha, \delta)$-sound for a class of MDPs $\mathcal{M}$ if it is $(\alpha, \delta)$-sound for every MDP in the class. The query complexity of a planner over $\mathcal{M}$ is defined as the maximum of its expected query cost over the members of the class.*

In the rest of the section, for $d \ge 1$ and $L > 0$, we use $\mathcal{B}_d(L) = \{x \in \mathbb{R}^d : \|x\| \le L\}$ to denote the $d$-dimensional Euclidean ball of radius $L$ centered at the origin.

## H.1 Exponential lower bound for planners with small suboptimality

We first show an exponential query complexity lower bound for sound planners that guarantee a small suboptimality bound. The result is a simple application of the techniques in Lattimore et al. [2020], and establishes the barrier for the suboptimality attainable by query-efficient planners:

**Theorem H.2.** *Let $\delta \le 0.9$, $\alpha \le 0.49/(1 - \gamma)$, and $\varepsilon \ge 0$, $d \ge 3$. There is a class of MDPs $\mathcal{M}$ with uniform policy value-function approximation error $\varepsilon$ for some $d$-dimensional feature map such that the query complexity of any $(\alpha, \delta)$-sound planner over $\mathcal{M}$ is at least $\exp\left(\Omega\left(d\left(\frac{\varepsilon}{\alpha(1-\gamma)}\right)^2\right)\right)$.*

*Proof.* Our proof is based on a similar complexity lower bound of Lattimore et al. [2020] for the multi-armed bandit setting, which is a special case of our problem. As such, we start by rewriting the class of bandit problems they used in their proof in our MDP framework, introducing a set of MDPs $\tilde{\mathcal{M}}$ each of which gets into a terminal state with no rewards after the first step. Let $\alpha' = 2.01\alpha(1 - \gamma) \le 1$ and $k = \left\lfloor \exp\left(\frac{d-2}{8}\left(\frac{\varepsilon}{\alpha'}\right)^2\right)\right\rfloor$. $\tilde{\mathcal{M}} = \{\tilde{M}_1, \ldots, \tilde{M}_k\}$ is defined to be a set of $k$ MDPs as follows: Each MDP in $\tilde{\mathcal{M}}$ has $k$ actions (i.e., $\mathcal{A} = [k]$) and two states: $\mathcal{S} = (s_0, s_1)$ with $s_0$ being the initial state, and deterministic transitions $P(s_1|s, a) = 1$ and $P(s_0|s, a) = 0$ for all $(s, a) \in \mathcal{S} \times \mathcal{A}$. For any $i \in [k]$, the reward distribution $\mathcal{R}_i$ for MDP $\tilde{M}_i$ is defined as follows: rewards for state $s_1$ are deterministically zero, that is, $\mathcal{R}_i(0|s_1, a) = 1$ for all $a \in \mathcal{A}$, making $s_1$ an absorbing state with zero reward, while rewards for state $s_0$ are deterministically $\alpha'$ for action $i$ and zero otherwise, that is, $\mathcal{R}_i(\alpha'|s_0, i) = 1$ and $\mathcal{R}_i(0|s_0, j) = 1$ for $j \in [\mathcal{A}]$ with $j \ne i$. Since this class of MDPs is equivalent to the class of muti-armed bandit problems defined by Lattimore et al. [2020], their proof of Corollary 3.3 implies that

- there exists a feature map $\tilde{\varphi} : \mathcal{S} \times \mathcal{A} \to \mathcal{B}_{d-1}(1)$ such that $\varepsilon$ is the maximum uniform policy value-function approximation error (Definition 1.1) over $\tilde{\mathcal{M}}$ equipped with features $\tilde{\varphi}$; and

- any planner that almost surely outputs an $\alpha'$-optimal *deterministic* policy for all $\tilde{M} \in \tilde{\mathcal{M}}$ (when run with a random access simulator for $\tilde{M}$) executes at least

$$\frac{1}{2}\exp\left(\frac{d-2}{8}\left(\frac{\varepsilon}{\alpha'}\right)^2\right) \tag{27}$$

queries in expectation.

We construct a new set $\mathcal{M} = \{M_1, \ldots, M_k\}$ of $k$ MDPs where for each $i \in [k]$, $M_i$ is a slight modification of $\tilde{M}_i$, always returning to the initial state $s_0$ instead of stopping after the first step: as such, the only modification is that the transition probabilities for all $M \in \mathcal{M}$ are $P(s_0|s, a) = 1$ and $P(s_1|s, a) = 0$ for all $(s, a) \in \mathcal{S} \times \mathcal{A}$. Let $\varphi : \mathcal{S} \times \mathcal{A} \to \mathcal{B}_d(2)$ be the features for all MDPs in $\mathcal{M}$, where for all $(s, a) \in \mathcal{S} \times \mathcal{A}$, $\varphi(s, a)$ is a concatenation of the $(d-1)$-dimensional $\tilde{\varphi}(s, a)$ and the scalar 1, so that the $d^{\text{th}}$ coordinate of $\varphi(s, a)$ is $\varphi(s, a)_d = 1$.

Fix any $i \in [k]$ and any stationary deterministic memoryless policy $\pi$, and let $\tilde{\theta}$ be the parameter realizing the low approximation error for $\tilde{M}_i$ and $\tilde{\varphi}$, that is, satisfying Eq. (19) (see Appendix C for a proof that such a $\tilde{\varphi}$ exists). In what follows, we denote $q$- and $v$-functions (with arbitrary superscripts) of an MDP $M$ by adding $M$ as a superscript to the corresponding function. Let $\theta$ be a concatenation of $\tilde{\theta}$ and the scalar $\gamma v_{M_i}^\pi(s_0)$. For any $(s, a) \in \mathcal{S} \times \mathcal{A}$,

$$q_{M_i}^\pi(s, a) = q_{\tilde{M}_i}^\pi(s, a) + \gamma v_{M_i}^\pi(s_0) \approx_\varepsilon \left\langle \tilde{\varphi}(s, a), \tilde{\theta} \right\rangle + \gamma v_{M_i}^\pi(s_0) = \left\langle \varphi(s, a), \theta \right\rangle .$$

The uniform policy value-function approximation error therefore remains at most $\varepsilon$ for $M_i$ with feature map $\varphi$, and this is true for any $i \in [k]$. We can therefore take any $(\alpha, \delta)$-sound planner with query complexity $T$ (for some $T \geq 0$) over $\mathcal{M}$, and provide it with a simulator of $M_i$ for any $i \in [k]$ (which we can trivially build with access to a simulator of $\tilde{M}_i$), to get a policy $\pi$ that is $\alpha$-optimal for $M_i$ with $\mathcal{P}_{M_i}$-probability at least $1 - \delta$. Recall that the rewards of $M_i$ are 0 for every action apart from a single optimal action, $i$, where the reward is $\alpha'$. Thus, $v_{M_i}^\star(s_0) = \alpha'/(1 - \gamma)$ and $v_{M_i}^\pi(s_0) = \alpha' \pi(i|s_0)/(1 - \gamma) = \pi(i|s_0) v_{M_i}^\star(s_0)$. Thus, with probability at least $1 - \delta$, $v_{M_i}^\star(s_0) - v_{M_i}^\pi(s_0) \leq \alpha < 0.5\alpha'/(1 - \gamma) = 0.5 v_{M_i}^\star(s_0)$. Therefore, $\pi(i|s_0) > 0.5$. As we know that the optimal action achieves a deterministic reward of $\alpha'$, we can test with a single query whether the action that $\pi$ assigns the highest probability to is optimal. If not, we can run the planner again and repeat the check. Since each run of the planner is successful with probability at least $1 - \delta$, independently of each other, almost surely one of the checks eventually passes and we output the deterministic policy that chooses the optimal action. Now the number of times the planner needs to be run is a stopping time (with respect to the sequence of the runs) with expectation at most $1/(1 - \delta)$, hence the expected query cost of the whole procedure is at most $(T + 1)/(1 - \delta)$ by Wald's equation. Note that the same policy is $\alpha'$-optimal for $\tilde{M}_i$. Therefore, the planner defined above almost surely outputs an $\alpha'$-optimal *deterministic* policy for any MDP in $\tilde{\mathcal{M}}$, and hence by Eq. (27) we have

$$T \geq \frac{1}{2}(1 - \delta) \exp\left(\frac{d-2}{8}\left(\frac{\varepsilon}{\alpha'}\right)^2\right) - 1 .$$

Therefore $T = \exp\left(\Omega\left(d\left(\frac{\varepsilon}{\alpha(1-\gamma)}\right)^2\right)\right)$, finishing the proof. $\qquad\square$

## H.2 Lower bound for linear MDPs

We close this section by proving a lower bounds on the query complexity of *random access* planners for linear MDPs (c.f. Theorem H.3).

We start by recalling the definition of linear MDPs [Zanette et al., 2020]: An MDP with countable state space is said to be *linear* if there exists a feature map $\varphi : \mathcal{S} \times \mathcal{A} \to \mathcal{B}_d(L)$, a state-transition feature map $\psi : \mathcal{S} \to \mathbb{R}^d$, and a reward parameter $\theta_r \in \mathcal{B}_d(B)$ such that $r(s, a) = \langle \varphi(s, a), \theta_r \rangle$ and $P(s'|s, a) = \langle \varphi(s, a), \psi(s') \rangle$ for any $(s, a, s') \in \mathcal{S} \times \mathcal{A} \times \mathcal{S}$, and $\sum_{s \in \mathcal{S}} \|\psi(s)\|_2 \leq B$. Clearly, any linear MDP satisfies Definition 1.1 with $\varepsilon = 0$. As such, the lower bounds presented below trivially transfer to the $\varepsilon$ uniform policy value-function approximation error case for any $\varepsilon \geq 0$.

**Theorem H.3.** *Let $\delta \in (0, 0.08]$, $\gamma \in [\frac{7}{12}, 1]$, $H = 1/(1 - \gamma)$, $\alpha \in (0, 0.05\gamma H/(1 + \gamma)^2]$, and $d \geq 3$. Then there is a class of linear MDPs $\mathcal{M}$ such that the query complexity of any $(\alpha, \delta)$-sound planner over $\mathcal{M}$ is at least $\Omega\left(d^2 H^3/\alpha^2\right)$.*

In the remainder of the section we prove the above bound. Throughout we assume that the conditions in Theorem H.3 are satisfied. We start with the construction of the class $\mathcal{M}$ of MDPs, then prove several auxiliary results, before finally presenting the proof of the theorem.

The construction of $\mathcal{M}$ is based on a combination of hard tabular MDPs [Xiao et al., 2022] and hard linear bandit problems [Lattimore and Szepesvári, 2020, Section 24.1]. Each MDP in $\mathcal{M}$ has two states: $\mathcal{S} = \{s_0, s_1\}$ with $s_0$ being the initial state. The action space is the intersection of a unit sphere

and a $(d-2)$-dimensional hypercube: $\mathcal{A} = \{\pm 1/\sqrt{d-2}\}^{d-2}$. We construct MDPs $M_\beta$ for all $\beta \in \mathcal{A}$, and let $\mathcal{M} = \{M_\beta \mid \beta \in \mathcal{A}\}$. The feature map $\varphi$ is defined, for any $a \in \mathcal{A}$, as

$$\varphi(s_0, a) = (1, 0, a^\top)^\top \quad \text{and} \quad \varphi(s_1, a) = (0, 1, 0, \ldots, 0)^\top.$$

We define the linear MDPs $M_\beta$ to have deterministic rewards for any $\beta \in \mathcal{A}$. Thus, $M_\beta$ is fully defined by its reward parameter $\theta_r$ and state-transition feature map $\psi$, according to the definition of linear MDPs. Let $\theta_r = (1, 0, \ldots, 0)^\top$, making state $s_0$ the only rewarding state, as then for all $a \in \mathcal{A}$,

$$r_\beta(s_0, a) = \langle \theta_r, \varphi(s_0, a) \rangle = 1 \quad \text{and} \quad r_\beta(s_1, a) = \langle \theta_r, \varphi(s_0, a) \rangle = 0.$$

Let $\Delta = 4(1 + \gamma)^2 \alpha / (\gamma H^2)$; since $\alpha \leq 0.05 \gamma H / (1 + \gamma)^2$, $\Delta \leq 0.2/H = 0.2(1 - \gamma)$. Let

$$\psi(s_0) = (\gamma, 0, \Delta\beta^\top)^\top \quad \text{and} \quad \psi(s_1) = (1 - \gamma, 1, -\Delta\beta^\top)^\top.$$

This implies that

$$P_\beta(s_0|s_0, a) = \gamma + \Delta\beta^\top a, \qquad\qquad P_\beta(s_1|s_0, a) = 1 - \gamma - \Delta\beta^\top a,$$
$$P_\beta(s_0|s_1, a) = 0, \qquad\qquad P_\beta(s_1|s_1, a) = 1.$$

Our assumptions guarantee that $P_\beta$ defines a valid transition kernel with probabilities in $[0, 1]$. The MDP starts in $s_0$ and rewards are collected until the state transitions to $s_1$, which is a terminal state with zero reward.

For the proof, we also need the following notation and supporting lemmas.

**Notation.** The probability measure $\mathcal{P}_{M_\beta}$ induced by the interconnection of a planner and a simulator for $M_\beta$ is written for simplicity as $\mathcal{P}_\beta$. Similarly, $\mathbb{E}_{M_\beta}$ is written as $\mathbb{E}_\beta$. $v_\beta$ (with arbitrary superscripts) denotes value functions (corresponding to the superscripts) of $M_\beta$. For any integer $i \in \{1, \ldots, d-2\}$, $\text{err}_i(\pi, \beta) = \sum_{a \in \mathcal{A}} \pi(a|s_0) I_{\text{sgn}(a_i) \neq \text{sgn}(\beta_i)}$ denotes the average error of a policy $\pi$ at the $i^{\text{th}}$ coordinate, where $a_i$ and $\beta_i$ are the $i^{\text{th}}$ components of $a$ and $\beta$, respectively, and $I_E$ is the indicator function of event $E$. With a slight abuse of notation, for a stationary memoryless policy $\pi$, we let $\pi^\top \beta$ denote $\sum_{a \in \mathcal{A}} \pi(a|s_0) a^\top \beta$.

**Lemma H.4.** *For any $M_\beta \in \mathcal{M}$, the value function of a stationary memoryless policy $\pi$ is given by*

$$v_\beta^\pi(s_0) = \frac{1}{1 - \gamma^2 - \gamma\Delta\pi^\top\beta}, \quad \text{and} \quad v_\beta^\pi(s_1) = 0.$$

*Proof.* It clearly holds that $v_\beta^\pi(s_1) = 0$. From the Bellman equation, $v_\beta^\pi(s_0) = 1 + \gamma(\gamma + \Delta\pi^\top\beta)v_\beta^\pi(s_0)$, and the claim follows from solving this equation for $v_\beta^\pi(s_0)$. $\qquad\square$

It is easy to see that the optimal policy in $M_\beta$ is defined by $\pi_\beta^\star(\beta|s_0) = 1$ (the actions in $s_1$ do not matter). Hence, by the above lemma,

$$v_\beta^\star(s_0) - v_\beta^\pi(s_0) = \frac{\gamma\Delta(1 - \pi^\top\beta)}{\left(1 - \gamma^2 - \gamma\Delta\right)\left(1 - \gamma^2 - \gamma\Delta\pi^\top\beta\right)}. \tag{28}$$

Because $1 - \pi^\top\beta = 2\sum_{i=1}^{d-2} \text{err}_i(\pi, \beta)/(d-2)$,

$$v_\beta^\star(s_0) - v_\beta^\pi(s_0) = \frac{2\gamma\Delta \sum_{i=1}^{d-2} \text{err}_i(\pi, \beta)}{(d-2)\left(1 - \gamma^2 - \gamma\Delta\right)\left(1 - \gamma^2 - \gamma\Delta\pi^\top\beta\right)}. \tag{29}$$

Accordingly, to prove a lower bound on the suboptimality of $\pi$, we need a lower bound for the sum of errors, $\sum_{i=1}^{d-2} \text{err}_i(\pi, \beta)$. To this end, Lemma H.5 below plays a key role.

**Lemma H.5** (Error Probability Lower Bound). *For any planner there exists a $\beta \in \mathcal{A}$ such that*

$$\sum_{i=1}^{d-2} \mathcal{P}_\beta\left(\text{err}_i(\pi_\tau, \beta) \geq \frac{1}{2}\right) \geq \frac{d-2}{2} - \frac{d-2}{2}\sqrt{1 - \exp\left(-\frac{5\Delta^2 H \mathbb{E}_\beta[\tau]}{(d-2)^2}\right)}. \tag{30}$$

To prove Lemma H.5, we need some technical lemmas. First, let $\mathcal{F}_t$ for any $t \in \mathbb{N}^+$ denote the $\sigma$-algebra generated by random variables in $H_t$, with $\mathcal{F}_1$ being the trivial $\sigma$-algebra. $\mathbb{F} = (\mathcal{F}_t)_{t=1}^{\infty}$ is chosen to be the filtration. The following lemma is adopted from Exercise 15.7 of Lattimore and Szepesvári [2020] with a slight modification.

**Lemma H.6** (KL-divergence decomposition). *Let $M$ and $M'$ be two MDPs differing only in their transition probability kernels, denoted by $P$ and $P'$, respectively. Then, for any any $\mathbb{F}$-adapted stopping time $\tau$ satisfying $\mathcal{P}_M (\tau < \infty) = 1$, and an $\mathcal{F}_\tau$-measurable[1] random variable $Z$,*

$$\mathrm{KL}\left(\mathcal{P}_M^Z \big\| \mathcal{P}_{M'}^Z\right) \leq \sum_{(s,a) \in \mathcal{S} \times \mathcal{A}} \mathbb{E}_M \left[\mathcal{N}_\tau(s,a)\right] \mathrm{KL}\left(P(\cdot|s,a)\|P'(\cdot|s,a)\right),$$

*where $\mathcal{P}_M^Z$ and $\mathcal{P}_{M'}^Z$ are the laws of $Z$ under $\mathcal{P}_M$ and $\mathcal{P}_{M'}$, respectively, $\mathcal{N}_t(s,a)$ denotes the number of queries with $(s,a) \in \mathcal{S} \times \mathcal{A}$ up to time step $t$, and $\mathrm{KL}(\cdot, \cdot)$ denotes the Kullback-Leibler (KL-) divergence of two distributions.*

The next lemma provides an upper bound on the KL-divergence of certain next-state distributions. A similar result appears in the proof of Lemma 6.8 of Zhou et al. [2020], but it requires that $\gamma \geq 2/3$; ours only requires the weaker assumption that $\gamma \geq 7/12$.

**Lemma H.7.** *Take any $\beta, \beta' \in \mathcal{A}$ that only differ at a single coordinate. Then for any action $a \in \mathcal{A}$,*

$$\mathrm{KL}\left(P_\beta(\cdot|s_0,a)\big\|P_{\beta'}(\cdot|s_0,a)\right) \leq \frac{5\Delta^2 H}{(d-2)^2}.$$

*Proof.* Our proof relies on Proposition 2 of Xiao et al. [2022]: for two Bernoulli distributions $\mathrm{Ber}(p)$ and $\mathrm{Ber}(p')$ with parameters $p, p' \in (0,1)$, it holds that

$$\mathrm{KL}\left(\mathrm{Ber}(p)\|\mathrm{Ber}(p')\right) \leq \frac{(p-p')^2}{2\min\{p(1-p), p'(1-p')\}}.$$

Since $P_\beta(s_1|s_0,a) = 1 - \gamma - \Delta\beta^\top a$ and $P_{\beta'}(s_1|s_0,a) = 1 - \gamma - \Delta(\beta')^\top a$,

$$\mathrm{KL}\left(P_\beta(\cdot|s_0,a)\big\|P_{\beta'}(\cdot|s_0,a))\right) \leq \frac{\Delta^2((\beta - \beta')^\top a)^2}{2\min_{b \in \mathcal{A}}(\gamma + \Delta\beta^\top b)(1 - \gamma - \Delta\beta^\top b)}$$

$$= \frac{2\Delta^2}{(d-2)^2 \min_{b \in \mathcal{A}}(\gamma + \Delta\beta^\top b)(1 - \gamma - \Delta\beta^\top b)} \tag{31}$$

for any action $a \in \mathcal{A}$. Note that

$$\min_{b \in \mathcal{A}}(\gamma + \Delta\beta^\top b)(1 - \gamma - \Delta\beta^\top b) \overset{(a)}{\geq} (\gamma + \Delta)(1 - \gamma - \Delta) \overset{(b)}{\geq} \frac{1 - \gamma - \Delta}{2} \overset{(c)}{\geq} \frac{2(1-\gamma)}{5},$$

where $(a)$ is due to the fact that $x(1-x)$ is monotone decreasing for $x \geq 0.5$ and $\gamma + \Delta\beta^\top b \geq \gamma - \Delta \geq 0.5$ since $\gamma \geq 7/12$ and $\Delta \leq 0.2(1 - \gamma)$, $(b)$ follows since $0.5 \leq \gamma + \Delta$, and $(c)$ holds because $\Delta \leq 0.2(1 - \gamma)$. Combining this result with Eq. (31) concludes the proof of the lemma. □

Now we are ready to prove Lemma H.5.

*Proof of Lemma H.5.* Let $\beta^{(i)}$ be a vector obtained by flipping the sign of $\beta$'s $i^{\text{th}}$ coordinate. Then,

$$\mathcal{P}_\beta\left(\mathrm{err}_i(\pi_\tau, \beta) \geq \frac{1}{2}\right) + \mathcal{P}_{\beta^{(i)}}\left(\mathrm{err}_i(\pi_\tau, \beta^{(i)}) \geq \frac{1}{2}\right)$$

$$= \mathcal{P}_\beta\left(\mathrm{err}_i(\pi_\tau, \beta) \geq \frac{1}{2}\right) + \mathcal{P}_{\beta^{(i)}}\left(\mathrm{err}_i(\pi_\tau, \beta) \leq \frac{1}{2}\right)$$

$$\geq \mathcal{P}_\beta\left(\mathrm{err}_i(\pi_\tau, \beta) \geq \frac{1}{2}\right) + \mathcal{P}_{\beta^{(i)}}\left(\mathrm{err}_i(\pi_\tau, \beta) < \frac{1}{2}\right)$$

$$\geq 1 - \sqrt{1 - \exp\left(-\mathrm{KL}\left(\mathcal{P}_\beta^{\mathrm{err}_i(\pi_\tau, \beta)}\big\|\mathcal{P}_{\beta^{(i)}}^{\mathrm{err}_i(\pi_\tau, \beta)}\right)\right)}$$

---

[1]By a slight abuse of notation, $\mathcal{F}_\tau$ is the $\sigma$-algebra generated by the random vector (with random length) $(S_1, A_1, R_1, S_1', \ldots, S_{\tau-1}, A_{\tau-1}, R_{\tau-1}, S_{\tau-1}')$.

where $\mathcal{P}_\beta^{\mathrm{err}_i(\pi_\tau,\beta)}, \mathcal{P}_{\beta^{(i)}}^{\mathrm{err}_i(\pi_\tau,\beta)} \in \mathcal{M}_1([0,1])$ are the laws of the random variable $\mathrm{err}_i(\pi_\tau,\beta)$ in $M_\beta$ and $M_{\beta^{(i)}}$, respectively , and the last line follows from an improved Bretagnolle-Huber inequality (inequality (14.11) of Lattimore and Szepesvári [2020]). Applying Lemmas H.6 and H.7 to the KL-divergence in the exponent in the right hand side of the above inequality together with the fact that $\sum_{(s,a)\in\mathcal{S}\times\mathcal{A}} \mathbb{E}_\beta\left[\mathcal{N}_\tau(s,a)\right] \le \mathbb{E}_\beta[\tau]$, we can further lower-bound the last line by

$$1 - \sqrt{1 - \exp\left(-\mathrm{KL}\left(\mathcal{P}_\beta^{\mathrm{err}_i(\pi_\tau,\beta)} \middle\| \mathcal{P}_{\beta^{(i)}}^{\mathrm{err}_i(\pi_\tau,\beta)}\right)\right)} \ge 1 - \sqrt{1 - \exp\left(-\frac{5\Delta^2 H \mathbb{E}_\beta[\tau]}{(d-2)^2}\right)}.$$

Therefore,

$$\frac{1}{|\mathcal{A}|} \sum_{\beta\in\mathcal{A}} \sum_{i=1}^{d-2} \mathcal{P}_\beta\left(\mathrm{err}_i(\pi_\tau,\beta) \ge \frac{1}{2}\right)$$

$$= \frac{1}{|\mathcal{A}|} \sum_{i=1}^{d-2} \frac{1}{2} \sum_{\beta\in\mathcal{A}} \left[\mathcal{P}_\beta\left(\mathrm{err}_i(\pi_\tau,\beta) \ge \frac{1}{2}\right) + \mathcal{P}_{\beta^{(i)}}\left(\mathrm{err}_i(\pi_\tau,\beta^{(i)}) \ge \frac{1}{2}\right)\right]$$

$$\ge \frac{d-2}{2} - \frac{d-2}{2}\sqrt{1 - \exp\left(-\frac{5\Delta^2 H \mathbb{E}_\beta[\tau]}{(d-2)^2}\right)}$$

where the first equality holds because for any $\beta$, there is exactly one $\beta^{(i)}$ in $\mathcal{A}$. As $\max_{\beta\in\mathcal{A}} f(\beta) \ge \sum_{\beta\in\mathcal{A}} f(\beta)/|\mathcal{A}|$ for any $f : \mathcal{A} \to \mathbb{R}$, $\arg\max_{\beta\in\mathcal{A}} \sum_{i=1}^{d-2} \mathcal{P}_\beta\left(\mathrm{err}_i(\pi_\tau,\beta^{(i)}) \ge 1/2\right)$ satisfies the claim of the lemma. $\qquad\square$

Now we are ready to prove Theorem H.3.

*Proof of Theorem H.3.* Take any $(\alpha,\delta)$-sound planner on $\mathcal{M}$. Let $\mathrm{err}(\pi,\beta) := \sum_{i=1}^{d-2} \mathrm{err}_i(\pi,\beta)$ for brevity. From Eq. (29),

$$\mathbb{E}_\beta\left[v_\beta^\star(s_0) - v_\beta^{\pi_\tau}(s_0)\right] \ge \frac{2\gamma\Delta\mathbb{E}_\beta\left[\mathrm{err}(\pi_\tau,\beta)\right]}{(d-2)\left(1 - \gamma^2 - \gamma\Delta\right)\left(1 - \gamma^2 + \gamma\Delta\right)} \tag{32}$$

$$\ge \frac{\gamma\Delta}{(d-2)\left(1 - \gamma^2 - \gamma\Delta\right)\left(1 - \gamma^2 + \gamma\Delta\right)} \sum_{i=1}^{d-2} \mathcal{P}_\beta\left(\mathrm{err}_i(\pi_\tau,\beta) \ge \frac{1}{2}\right)$$

$$\ge \frac{\gamma\Delta}{2\left(1 - \gamma^2 - \gamma\Delta\right)\left(1 - \gamma^2 + \gamma\Delta\right)}\left(1 - \sqrt{1 - \exp\left(-\frac{5\Delta^2 H \mathbb{E}_\beta[\tau]}{(d-2)^2}\right)}\right), \tag{33}$$

where the first inequality holds because $\pi^\top\beta \ge -1$ for any stationary memoryless policy $\pi$, the second inequality is due to the Markov inequality, while the last inequality holds by Lemma H.5. From Eq. (29) and $\pi^\top\beta \le 1$ we also have that

$$\mathbb{E}_\beta\left[v_\beta^\star(s_0) - v_\beta^{\pi_\tau}(s_0)\right] \le \frac{2\gamma\Delta\mathbb{E}_\beta\left[\mathrm{err}(\pi_\tau,\beta)\right]}{(d-2)\left(1 - \gamma^2 - \gamma\Delta\right)^2}$$

$$\le \frac{\gamma\Delta}{4\left(1 - \gamma^2 - \gamma\Delta\right)^2}\left[7\mathcal{P}_\beta\left(\mathrm{err}(\pi_\tau,\beta) > \frac{d-2}{8}\right) + 1\right],$$

where the second inequality holds because

$$\mathbb{E}_\beta\left[\mathrm{err}(\pi_\tau,\beta)\right] = \mathbb{E}_\beta\left[\mathrm{err}(\pi_\tau,\beta)I_{\mathrm{err}(\pi_\tau,\beta)>\frac{d-2}{8}} + \mathrm{err}(\pi_\tau,\beta)I_{\mathrm{err}(\pi_\tau,\beta)\le\frac{d-2}{8}}\right]$$

$$\le \mathbb{E}_\beta\left((d-2)I_{\mathrm{err}(\pi_\tau,\beta)>\frac{d-2}{8}} + \frac{d-2}{8}I_{\mathrm{err}(\pi_\tau,\beta)\le\frac{d-2}{8}}\right)$$

$$= \frac{d-2}{8}\left(7\mathcal{P}_\beta\left(\mathrm{err}(\pi_\tau,\beta) > \frac{d-2}{8}\right) + 1\right).$$

Combining this result with Eq. (33),

$$\mathcal{P}_\beta\left(\text{err}(\pi_\tau,\beta) > \frac{d-2}{8}\right) \geq \frac{2}{7}\frac{1-\gamma^2-\gamma\Delta}{1-\gamma^2+\gamma\Delta}\left(1-\sqrt{1-\exp\left(-\frac{5\Delta^2 H\mathbb{E}_\beta[\tau]}{(d-2)^2}\right)}\right) - \frac{1}{7}$$

$$= \frac{2}{7}\left(1-\frac{2\gamma\Delta}{1-\gamma^2+\gamma\Delta}\right)\left(1-\sqrt{1-\exp\left(-\frac{5\Delta^2 H\mathbb{E}_\beta[\tau]}{(d-2)^2}\right)}\right) - \frac{1}{7}$$

$$> \frac{2}{7}\left(1-\frac{2\gamma\Delta}{1-\gamma^2}\right)\left(1-\sqrt{1-\exp\left(-\frac{5\Delta^2 H\mathbb{E}_\beta[\tau]}{(d-2)^2}\right)}\right) - \frac{1}{7}.$$

Note that $\text{err}(\pi_\tau,\beta) > (d-2)/8$ implies that $v_\beta^\star(s_0) - v_\beta^{\pi_\tau}(s_0) > \alpha$ since similarly to Eq. (32) (i.e., without the expectation)

$$v_\beta^\star(s_0) - v_\beta^{\pi_\tau}(s_0) \geq \frac{2\gamma\Delta\,\text{err}(\pi_\tau,\beta)}{(d-2)\left((1-\gamma^2)^2-\gamma^2\Delta^2\right)} > \frac{1}{4}\frac{\gamma\Delta}{(1-\gamma^2)^2-\gamma^2\Delta^2} > \frac{1}{4}\frac{\gamma\Delta}{(1-\gamma^2)^2} = \alpha,$$

where the last equality follows because $\Delta = 4(1+\gamma)^2\alpha/(\gamma H^2) = 4(1-\gamma^2)^2\alpha/\gamma$. Therefore,

$$\mathcal{P}_\beta\left(v_\beta^\star(s_0) - v_\beta^{\pi_\tau}(s_0) > \alpha\right) \geq \mathcal{P}_\beta\left(\text{err}(\pi_\tau,\beta) > \frac{d-2}{8}\right)$$

$$> \frac{2}{7}\left(1-\frac{2\gamma\Delta}{1-\gamma^2}\right)\left(1-\sqrt{1-\exp\left(-\frac{5\Delta^2 H\mathbb{E}_\beta[\tau]}{(d-2)^2}\right)}\right) - \frac{1}{7}$$

$$\overset{(a)}{\geq} \frac{2}{7}\left(1-\frac{0.4\gamma}{1+\gamma}\right)\left(1-\sqrt{1-\exp\left(-\frac{5\Delta^2 H\mathbb{E}_\beta[\tau]}{(d-2)^2}\right)}\right) - \frac{1}{7}$$

$$\overset{(b)}{\geq} \frac{8}{35}\left(1-\sqrt{1-\exp\left(-\frac{5\Delta^2 H\mathbb{E}_\beta[\tau]}{(d-2)^2}\right)}\right) - \frac{1}{7}$$

$$= \frac{3}{35} - \frac{8}{35}\sqrt{1-\exp\left(-\frac{5\Delta^2 H\mathbb{E}_\beta[\tau]}{(d-2)^2}\right)}.$$

where $(a)$ follows since $\Delta \leq 0.2(1-\gamma)$, and $(b)$ follows since $0 \leq 0.4x/(1+x) \leq 0.2$ for $x \in [0,1]$. This implies that unless $\mathbb{E}_\beta[\tau] \geq \Omega\left(d^2 H^3/\alpha^2\right)$, the algorithm is not $(\alpha,\delta)$-sound. Indeed if

$$\mathbb{E}_\beta[\tau] \leq \frac{(d-2)^2}{5\Delta^2 H}\log\left(\frac{1}{1-\frac{(3-35\delta)^2}{64}}\right),$$

it holds that $\mathcal{P}_\beta\left(v_\beta^\star(s_0) - v_\beta^{\pi_\tau}(s_0) > \alpha\right) > \delta$, contradicting the assumption that the planner is $(\alpha,\delta)$-sound on $\mathcal{M}$ (the upper bound $\delta \leq 0.08 < 3/35$ guarantees that the logarithmic term above is bounded by a constant). This concludes the proof. □