# OpenReview forum: "Confident Approximate Policy Iteration for Efficient Local Planning in $q^\pi$-realizable MDPs"
_NeurIPS.cc/2022/Conference — NeurIPS 2022 Accept_

### Official Review · Reviewer_eM8M · 2022-07-11

**Rating:** 6
**Confidence:** 3
**Soundness:** 4 excellent
**Presentation:** 4 excellent
**Contribution:** 3 good

**Summary:**

The authors consider the linear action value function setting, and in particular the local planning problem with value-based methods.
They obtained an algorithm whose approximation error grows linearly with the horizon all the while returning a stationary policy. In terms of query complexity, they achieve the best known accuracy and query complexity at the same time.

**Questions:**

See above

**Limitations:**

The work is quite technical and theoretical and could be of interesting to a more narrow audience. Since the work seems to be computationally implementable, it would be nice to have some limited experiments as well (in simulated domains), although this is not standard for this sub-area of RL

**Strengths And Weaknesses:**

I believe that this is not the first result with an approximation error that scales linearly with the horizon. For example, see what could be inferred from “Q* Approximation Schemes for Batch Reinforcement Learning: A Theoretical Comparison” by Xi et al. (To be clear this work is quite different to the proposal here and does not diminish the novelty of the work submitted).

I believe that although the work strictly improve on prior art, it is nonetheless quite technical and of potential interest to a technical audience.

Question: this work revolves around improving the dependence on the approximation error. However, the definition of approximation error seems is very strong (\ell-infty). In practice, such value is likely to be large. Given that a local access is assumed, can it be relaxed to other more forgiving definitions? In particular, I am referring to non \ell-\infty definitions.

---

> ### Author Response · Authors · 2022-08-02
> **Regarding experiments**
>
> We agree that including experiments would be nice, but given that our paper is already quite long, we prefer to leave this to future work.

---

> ### Author Response · Authors · 2022-08-02
> **Regarding \ell-\infty approximation error**
>
> While assuming a low error may feel strong, we remark that this is a strictly weaker assumption than linear MDPs or approximately-linear MDPs (see Lemma C.1. of Chi et al. https://arxiv.org/abs/1907.05388 and Proposition 4 of Zanette et al., 2020, https://arxiv.org/abs/2003.00153), which also include block MDPs (among other classes). It is unknown to us whether a low uniform approximation error bound is enough for sample-efficient RL with online access, the local planner is the most challenging setting we are aware of where we have seen this to lead to polynomial bounds. However, we think finding a more permissive measure of approximation error is very important future work. While the \ell-\infty measure may look conservative, in general, dependence on this measure is unavoidable as shown in previous works (e.g. Lattimore et al. 2020). See also the answer to Reviewer 2pLv (https://openreview.net/forum?id=Q_WPshXgGI9&noteId=WGJJrQZC1P): The \ell-\infty error inflated even by a factor of $\sqrt{d}H$ is unavoidable.

---

> ### Author Response · Authors · 2022-08-02
> **Regarding Xie et al’s work**
>
> While this work is for the batch RL setting, we thank the reviewer for pointing out this connection. We certainly did not intend to suggest that our work is the first where the suboptimality scales with the horizon, as we point out in Table 1, even for our exact setting, Confident MC-POLITEX achieves the same suboptimality bound as our method, albeit at a much larger query cost. Our work is only the first at connecting this suboptimality with the best query cost available in the same algorithm.

---

### Official Review · Reviewer_UY7n · 2022-07-12

**Rating:** 6
**Confidence:** 4
**Soundness:** 3 good
**Presentation:** 3 good
**Contribution:** 2 fair

**Summary:**

The paper presents a confidence approximate policy iteration algorithm for discounted MDPs. The authors then prove that the algorithm produces a deterministic stationary policy with an optimal error bound which scales linearly with product of the effective horizon and the worst-case approximation error of the action-value functions of stationary policies.

**Questions:**

Algorithm and proof techniques very similar to existing works, the contribution is not very clear

**Ethics Review Area:**

["Responsible Research Practice (e.g., IRB, documentation, research ethics)"]

**Limitations:**

Proof techniques very similar to existing works

**Strengths And Weaknesses:**

The authors extend previous works to improve on the performance bounds but it is very similar to previous works by Yin et. al.

---

> ### Author Response · Authors · 2022-08-02
> **Summary of contributions and novel proof techniques**
>
> As with each paper dealing with $q^\pi$ realizability or linear MDPs, we use the tools and techniques common to linear function approximation (optimal design, core-set, uncertainties derived from Gram matrix) and the standard analysis of approximate policy iteration (API).
> However (as also summarized in the abstract), we make two contributions in this paper, which – in our opinion – should be of substantial interest to the community: (i) we introduce an innovative and intuitive variant of API (called Confident API – CAPI), which fixes in a simple way the issue that standard API faces, namely, scaling poorly with the misspecification error, while being more transparent than previous “fixes” and still producing a single stationary policy; and (ii) we provide a local planner for $q^\pi$-realizable MDPs with better performance bounds than SOTA.
>
> (i) It is known that in certain MDPs API does not achieve the optimal convergence rate (see Figure 1 of https://arxiv.org/abs/2007.11684 for an example). Our new algorithm resolves this problem (via the introduction of an extra decision step about when to update the policy in a given state). The benefit of this method (compared to that of Scherrer and Lesner [2012]) is that our method gives a stationary policy, which is important for our second contribution (as explained in detail in the paper in the discussion following Theorem 1.3).
>
> (ii) The improved bounds in our second contribution are due to two important innovations compared to Yin et al. [2022]: we apply CAPI instead of API and we get rid of the restarts present in Yin et al’s method. The latter is possible because of a new proof technique that provides uniform approximation error guarantees over a (possibly infinite) set of policies in the Extend(.,.) class, and the fact that we show that CAPI supports partial updates such that optimality guarantees achieved for any state are never lost. We strongly disagree that these proof techniques (especially the guarantees for the Extend class) are minor variations of existing techniques.

---

### Official Review · Reviewer_2pLv · 2022-07-20

**Rating:** 7
**Confidence:** 4
**Soundness:** 3 good
**Presentation:** 3 good
**Contribution:** 3 good

**Summary:**

This paper proposed a new algorithm in the setting of MDP with approximate linear value function (for any policy, the value function is approximately linear in l infinity norm with respect to a given feature map) along with a local access simulator. The algorithm CAPI achieves the best known sub-optimality bound along with the best known query cost in the local access setting, previous algorithms achieve only one of them. The algorithm is based on Policy iteration and a new technique of partition the observed state space into layers and planning policies for each layer with a refined accuracies, which maybe of independent interest.

**Questions:**

(1) Is there a lower bound in the random access setting, how does the result compare to it?

(2) Is it possible to use this idea in the general function approximation setting?

(3) What can we do if we are not granted with the feature but only have a feature class?

**Limitations:**

Yes

**Strengths And Weaknesses:**

Strengths: The algorithm is highly novel although the key idea of core set is proposed in previous papers. The proof is clean and looks sound to me. The result is pretty great since the algorithm achieve both the best known query complexity and accuracy. (Compared to [Lattimoore et al., 2020] and [Yin et al., 2022])

Weaknesses: Lack a lower bound. A more detailed discussion of why previous algorithm [Yin et al., 2022] cannot achieve the tight bound is recommended. As well as a comparison of computation complexity with previous methods.

---

> ### Author Response · Authors · 2022-08-02
> **Question 1**
>
> Q1: There are lower bounds, partly published in the literature. All of these work both in the random access and (of course) the local access setting. In addition, since the submission of the work, we discovered the following improvements on the state of the art:
> (a) We have discovered a simple lower bound showing that our suboptimality guarantee is optimal (to be included in the final version): Lattimore et al. [2020] provides a construction for the lower bound in the linear bandit case (Corollary 3.3 in their paper). We can easily extend this to an MDP that has exactly one state and after taking any action the MDP always remains in the same state. The actions in this MDP correspond to the bandit arms in Lattimore et al’s construction, but we append a scalar 1 to the feature vector of each action. It is easy to see that the resulting MDP is $q^\pi$-realizable with approximation error $\epsilon$ (using $\theta_1,...,\theta_d$ as in Lattimore et al’s construction, and defining $\theta_{d+1}=\gamma v^\pi(s_0)$).
> (b) We have also seen a (not yet published, under review) lower bound that implies that our query cost guarantee is sharp in terms of d and epsilon, but the query lower bound in terms of $H=1/(1-\gamma)$ remains an open question. We will include a reference to this work in the final version.

---

> ### Author Response · Authors · 2022-08-02
> **Question 2**
>
> Q2: We expect that our ideas could be extended to the case of nonlinear function approximation: CAPI (Section 3) requires no change (the performance bounds hold as long as we can control the estimation error). We expect that the rest of the ideas for CAPI-Qpi-Plan will transfer too, and the resulting bound will depend on the eluder dimension of the function class. As usual with nonlinear function approximation, further work will be needed to derive practical versions of this “general” algorithm. We expect the main challenge to be bounding the eluder dimension, as well as efficiently computing uncertainty bounds and good enough estimates of q functions.

---

> ### Author Response · Authors · 2022-08-02
> **Question 3**
>
> Q3: We are not sure we understand the question. Our algorithm only needs to know the feature class in advance (defined by the norm bound L), the feature vectors of a state are only revealed when a state is visited. Could you please clarify if you meant something else?
> If what you meant is that in every state we are only given a feature class, then please confirm this and we will be happy to sketch out an argument for why the problem can be hopelessly hard.

---

> ### Author Response · Authors · 2022-08-02
> **Comparison of compute and memory cost**
>
> We will clarify in the final version how our compute and memory bounds compare to Yin et al. The summary is that (a) their algorithms take $Ad^2$ computation per query; consulting Table 1 tells us that no algorithm’s compute cost dominates the other; (b) the memory cost overhead of CAPI-Qpi-Plan is a factor $H$ (where $H=1/(1-\gamma)$) over Yin’s algorithm; (c) the compute cost overhead of running our algorithm’s output policy is a factor $d^2 H$, and (d) the memory cost of our algorithm’s output policy is a factor $dH$ larger than Confident MC-LSPI, and no clear comparison to MC-POLITEX is possible as this algorithm’s memory use has a term that scales with $1/\epsilon^2 H^2$.
>
> In the final version, we will include these comparisons and outline open questions related to them.

---

> ### Author Response · Authors · 2022-08-02
> **Why Yin et al. doesn’t achieve the tight bounds**
>
> The tight query complexity bound is missed by Yin et al.'s algorithms because they restart every time a new core-set member is discovered, whereas our analysis allows us to never restart. The tight suboptimality bound is missed for LSPI because this is an inherent limitation of standard API (see Figure 1 of https://arxiv.org/abs/2007.11684 for an example). We plan to include a discussion of this as well in the final version.

---

### Meta-Review · Area_Chair_uvEF · 2022-08-23

**Recommendation:** Accept
**Confidence:** Certain

**Metareview:**

All reviewers and the AC believe this paper makes valuable contributions to the theoretical reinforcement learning community.

**Award:**

No

---

### Decision · Program_Chairs · 2022-09-14

Accept